# Characterization of the endogenous DAF-12 ligand and its use as an anthelmintic agent in *Strongyloides stercoralis*

Zhu Wang[1], Mi Cheong Cheong[1], Jet Tsien[2], Heping Deng[2], Tian Qin[2], Jonathan DC Stoltzfus[3], Tegegn G Jaleta[4], Xinshe Li[4], James B Lok[4], Steven A Kliewer[1,5]*, David J Mangelsdorf[1,6]*

[1]Department of Pharmacology, University of Texas Southwestern Medical Center, Dallas, United States; [2]Department of Biochemistry, University of Texas Southwestern Medical Center, Dallas, United States; [3]Department of Biology, Millersville University of Pennsylvania, Millersville, United States; [4]Department of Pathobiology, School of Veterinary Medicine, University of Pennsylvania, Philadelphia, United States; [5]Department of Molecular Biology, University of Texas Southwestern Medical Center, Dallas, United States; [6]Howard Hughes Medical Institute, University of Texas Southwestern Medical Center, Dallas, United States

**Abstract** A prevalent feature of *Strongyloides stercoralis* is a life-long and potentially lethal infection that is due to the nematode parasite's ability to autoinfect and, thereby, self-replicate within its host. Here, we investigated the role of the parasite's nuclear receptor, *Ss*-DAF-12, in governing infection. We identified Δ7-DA as the endogenous *Ss*-DAF-12 ligand and elucidated the hormone's biosynthetic pathway. Genetic loss of function of the ligand's rate-limiting enzyme demonstrated that Δ7-DA synthesis is necessary for parasite reproduction, whereas its absence is required for the development of infectious larvae. Availability of the ligand permits *Ss*-DAF-12 to function as an on/off switch governing autoinfection, making it vulnerable to therapeutic intervention. In a preclinical model of hyperinfection, pharmacologic activation of DAF-12 suppressed autoinfection and markedly reduced lethality. Moreover, when Δ7-DA was administered with ivermectin, the current but limited drug of choice for treating strongyloidiasis, the combinatorial effects of the two drugs resulted in a near cure of the disease.

*For correspondence:
steven.kliewer@utsouthwestern.edu (SAK);
davo.mango@utsouthwestern.edu (DJM)

## Editor's evaluation

This work reveals the pathway by which an important human parasite synthesizes a nuclear hormone receptor ligand critical for progression through its life cycle and demonstrates the potential therapeutic implications of perturbing this pathway. The experiments are insightfully and expertly conceived, designed and executed, and the data support the conclusions. This manuscript will be of general interest to parasitologists, nematode biologists, and those studying transcriptional regulatory networks governed by ligand-gated nuclear receptors.

## Introduction

Strongyloidiasis is a neglected tropical disease caused by the nematode parasite, *Strongyloides stercoralis*. It is estimated to infect ~600 million people worldwide and is endemic in Africa, Asia, Latin America, and parts of the Caribbean, southern United States and Europe (*Buonfrate et al., 2020*). Because of the unique nature of its lifecycle, *S. stercoralis* infections are often life-long and

**Figure 1.** Lifecycle of *S. stercoralis*. Similar to other nematodes, *S. stercoralis* hatch from eggs and undergo four larval (L) molts to become adults, either in the environment or in the host. The parasite has two infectious stages (highlighted in yellow), one that exists in the environment as L3i and one that exists in the mucosa of the host intestine as an autoinfective L3a. In the postparasitic (PP) environment, larvae can undergo two developmental fates. Under host-like temperature conditions, the PP-larvae arrest their development directly as infectious third-stage larvae. However, under more temperate conditions, the postparasitic L1 (PP-L1) develop through one free-living (FL) generation (green arrows), followed by a post-free-living (PFL) generation (red arrows) that invariably arrests as L3i. Pharmacologically activating the nuclear receptor DAF-12 has been shown to prevent L3i arrest in the PFL generation (*Albarqi et al., 2016*; *Wang et al., 2009*). In this study, we characterized the role of the endogenous DAF-12 ligand at each of these developmental stages. See text for details.

up to 2.5% of these infections will progress to a hyperinfection syndrome that has a 90% mortality rate if untreated (*Milder et al., 1981*). Notably, such hyperinfections are often caused by administering glucocorticoids to otherwise asymptomatic patients (*Milder et al., 1981*). The lifecycle of *S. stercoralis* is atypical of most soil-transmitted helminths (*Krolewiecki et al., 2013*; *Lok, 2007*; *Page et al., 2018*; *Figure 1*). Infective third-stage larvae (L3i) are developmentally quiescent and reside in fecal-contaminated soil until they contact the host and penetrate the skin, a process that reactivates their development as L3+ larvae. In the host, the activated L3+ immediately begin feeding, and typically migrate to the lungs, enter the alimentary canal, and transit to the intestine where the stage four larvae (L4) mature into parasitic (P)-adult females (there are no parasitic males). The P-adult females reproduce asexually and generate rhabditiform, noninfectious stage one larvae (L1) progeny that can enter two alternate lifecycle routes, one in the external environment and one in the host. Post-parasitic L1 (PP-L1) enter the external environment by leaving the host with the feces. Under conditions that mimic the host intestine (e.g., 37°C), most PP-L1 develop as females directly into infectious filariform L3i (*Albarqi et al., 2016*). However, under favorable conditions (e.g., 22 °C, high humidity) the majority of PP-L1 undergo a developmental switch and become free-living (FL) male and female larvae that eventually mature into FL-adults, which are morphologically distinct from parasitic females and feed on soil microbes. The FL-adults mate and produce post-free-living (PFL) larvae that are exclusively female and now definitively committed to become infective L3i larvae.

In addition to the conditions that produce L3i outside of the host, *S. stercoralis* has the ability to remain in the host and undergo continuous auto-infection (*Figure 1*). This is due to the presence of L1 larvae that develop directly into autoinfective third-stage larvae (L3a) within the intestine. Similar to their L3i counterparts, L3a infect the host, in this case by penetrating the intestinal wall and entering parenteral tissues as activated L3+ to complete the parasitic lifecycle. Autoinfection is unique to *S. stercoralis* and allows the parasite to persist in the host for decades as a latent, often asymptomatic infection. However, under certain host conditions such as immunosuppression by glucocorticoids, the autoinfection spirals out of control, dramatically increasing the parasite burden. The resulting hyper-infection leads to a breakdown of the intestinal mucosal barrier and dissemination of invasive larvae throughout the body accompanied by bacterial infection leading to sepsis that is responsible for the high mortality. The current treatment of choice is the nematode-selective chloride channel activator ivermectin, which is effective at controlling acute infections, but is limited by its inability to eradicate the persistent autoinfective larvae (*Krolewiecki et al., 2013*; *Repetto et al., 2018*). Perhaps for this reason, the efficacy of ivermectin in preventing fatality due to the hyperinfection and disseminated forms of the disease is as low as 50% (*Buonfrate et al., 2013*), and cases of ivermectin resistance are

now being reported in other nematode parasites (*Prichard, 2007*). The current increased prevalence of strongyloidiasis has stimulated efforts to include increased dosing of ivermectin through mass drug administration (*Bisoffi et al., 2013*), which in the long-term could accelerate ivermectin resistance in *S. stercoralis*. Of further immediate concern, the risk of hyperinfection has risen dramatically with the use of glucocorticoids to treat inflammatory diseases, particularly SARS-CoV-2 (*Moloo, 2020*). For these reasons, there is an urgent need to develop new therapeutic strategies for treating strongyloidiasis (*Krolewiecki et al., 2013*; *Moloo, 2020*).

The lifecycle of *S. stercoralis* has similarities to the free-living nematode, *C. elegans*. The developmentally arrested L3i stage of *S. stercoralis* is analogous to the L3 dauer (d) stage of *C. elegans*. In *C. elegans*, L3d development is governed by DAF-12 (*Antebi et al., 1998*; *Antebi et al., 2000*), a nematode-specific nuclear receptor that is conserved in parasitic species (*Ayoade et al., 2020*; *Long et al., 2020*; *Ma et al., 2019*; *Motola et al., 2006*; *Ogawa et al., 2009*; *Wang et al., 2017*; *Wang et al., 2009*). In favorable environments, developing larvae synthesize the endogenous *C. elegans* (*Ce*)-DAF-12 ligands called dafachronic acids (*Mahanti et al., 2014*; *Motola et al., 2006*). The most potent and abundant of these ligands is Δ7-dafachronic acid (Δ7-DA), which induces a transcriptional program that governs metabolism and reproductive growth to mature adults (*Bethke et al., 2009*; *Hammell et al., 2009*; *Wang et al., 2015*). The final and rate-limiting step of Δ7-DA synthesis is catalyzed by DAF-9, a cytochrome P450 (*Gerisch and Antebi, 2004*; *Jia et al., 2002*; *Motola et al., 2006*). In unfavorable environments, Δ7-DA is not produced and unliganded *Ce*-DAF-12 functions as a transcriptional repressor that arrests growth as L3d larvae in a process known as dauer diapause. When favorable conditions return, Δ7-DA synthesis resumes, DAF-12 is transcriptionally active, and worms exit L3 dauer and continue reproductive development. Our previous work demonstrated a similar requirement for the transactivation of the orthologous *S. stercoralis* (*Ss*)-DAF-12 receptor in governing L3i (*Albarqi et al., 2016*; *Wang et al., 2009*) and the essentiality of the receptor in this process (*Cheong et al., 2021*). Although the endogenous *Ss*-DAF-12 ligand was unknown, we demonstrated that pharmacologic administration of Δ7-DA to PP-L1 or PFL-L1 larvae commits them to FL reproductive development, even under conditions that would normally commit them to become L3i (*Albarqi et al., 2016*; *Wang et al., 2009*), and institutes a transcriptomic profile in L3i similar to L3+ (*Stoltzfus et al., 2014*). In addition, knockout of *Ss*-DAF-12 prevents the formation of L3i larvae (*Cheong et al., 2021*), and in a mouse model of hyperinfection, exogenous Δ7-DA treatment lowers the worm burden in the intestine (*Patton et al., 2018*).

Despite these findings, an understanding of the DAF-12 signaling pathway during parasitism and importantly whether targeting the *Ss*-DAF-12 receptor is a viable, curative strategy for treating strongylodiasis is lacking. In this report, we identified the endogenous *Ss*-DAF-12 ligand, characterized its biosynthetic pathway, and demonstrated how its regulation governs the parasite's lifecycle. Finally, using a relevant preclinical animal model, we show that pharmacological activation of *Ss*-DAF-12 overcomes the deficiencies of ivermectin, and that when used in combination with ivermectin, provides a combinatorial therapeutic response that may be curative.

## Results
### Δ7-DA is the endogenous ligand for DAF-12 in *S. stercoralis*
Our previous work had demonstrated Δ7-DA is able to bind and activate *Ss*-DAF-12 (*Wang et al., 2009*). However, whether Δ7-DA or a related molecule is the endogenous *S. stercoralis* ligand is unknown. Indeed, the finding that the ligand binding pocket of *Ss*-DAF-12 is divergent from that of *Ce*-DAF-12 (*Wang et al., 2009*) and the observation that *S. stercoralis* lacks an obvious ortholog of DAF-9 (the only known DA-synthesizing enzyme found in *C. elegans*) suggested that the *Ss*-DAF-12 ligand would likewise be divergent. To determine the identity of *S. stercoralis* DAF-12 ligands, we employed an unbiased, activity-based biochemical purification strategy (*Figure 2—figure supplement 1*). In a control experiment using a DA-deficient strain of *C. elegans* supplemented with a known quantity of DA, we demonstrated this strategy effectively recovered 70% of the ligand (*Figure 2—figure supplement 1B*). Following this strategy, we fractionated lipids extracted from the FL-L3 larvae of *S. stercoralis*, which are analogous to the L3 stage of *C. elegans* where DA levels are highest (*Li et al., 2013*; *Motola et al., 2006*). Among the 70 high-performance liquid chromatography (HPLC) fractions, *Ss*-DAF-12 ligand activity was detected only in fraction 23 in a cell-based reporter assay and

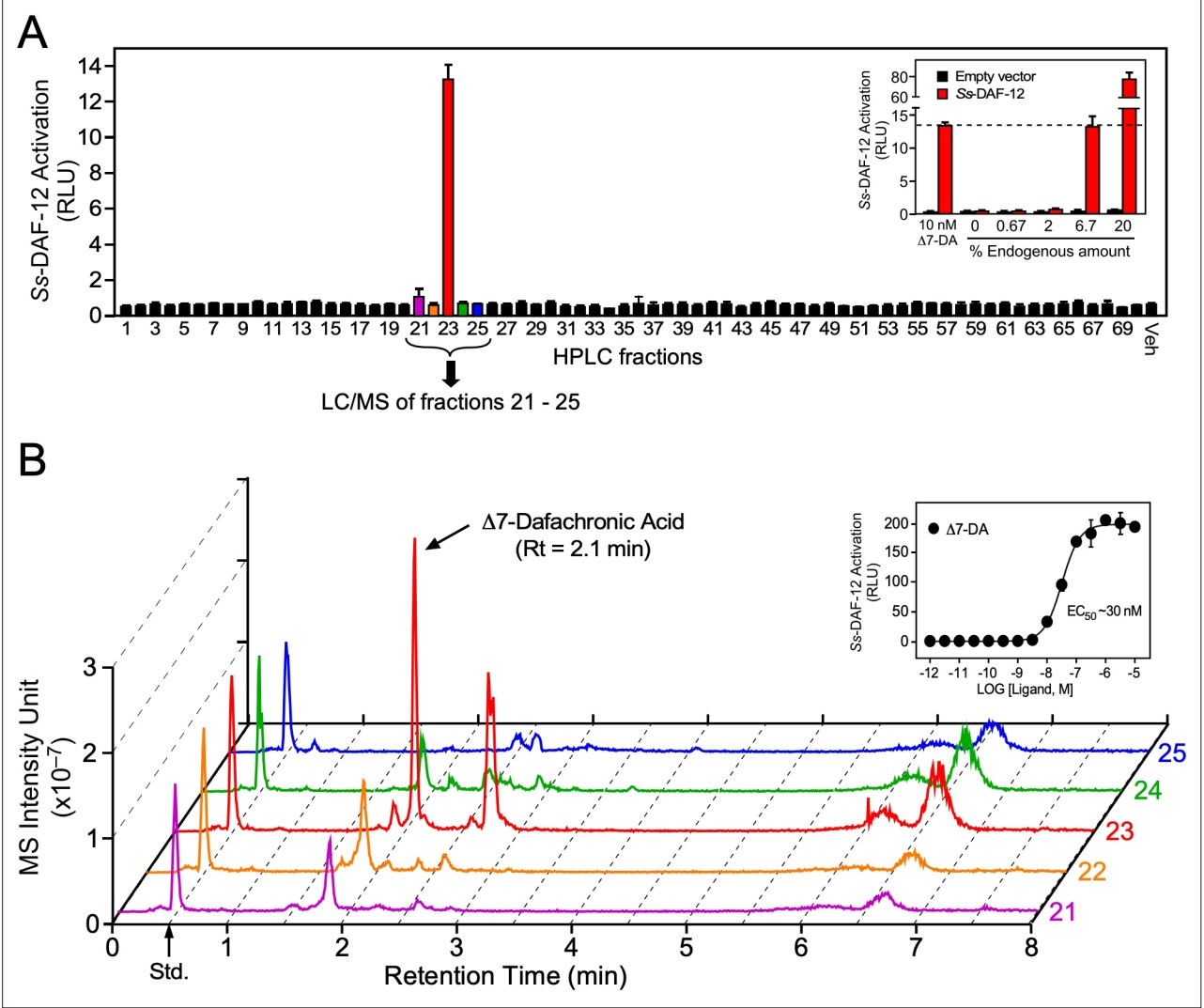

**Figure 2.** Identification of Δ7-dafachronic acid as the endogenous DAF-12 ligand in *S. stercoralis*. (**A**) Purification of the endogenous *S. stercoralis* ligand for *Ss*-DAF-12. Lipids from free-living L3 worms were extracted and fractionated as described in *Figure 2—figure supplement 1*. The resulting lipid fractions were then tested in a *Ss*-DAF-12 cell-based reporter assay. Inset: Dose response of the endogenous activity in fraction 23. RLUs, relative light units. Data are presented as the mean ± standard deviation (SD) of technical triplicates. (**B**) Δ7-DA is specifically present in the active lipid fraction. High-performance liquid chromatography (HPLC) fractions 21–25 were analyzed by ultra-performance liquid chromatography coupled with mass spectrometry (UPLC–MS). CDCA-$^2$H$_4$ (100 nM) was added to each fraction as an internal standard and has a retention time (RT) of 0.5 min (arrow). Inset: Dose response of Δ7-DA in the *Ss*-DAF-12 reporter assay. Data are presented as the mean ± SD of technical triplicates and were repeated three times. See also *Figure 2—figure supplements 1 and 2*.

The online version of this article includes the following figure supplement(s) for figure 2:

**Figure supplement 1.** Strategy for activity-based, DAF-12 ligand purification in *S. stercoralis*.

**Figure supplement 2.** Δ4-DA and Δ1,7-DA are not present in free-living L3 parasites.

this activity was dose dependent (*Figure 2A*). Remarkably, fraction 23 coeluted with dafachronic acids (*Figure 2—figure supplement 1B*), suggesting that the ligand is a related molecule.

To test whether the identified ligand activity is a dafachronic acid, we utilized a method incorporating ultra-performance liquid chromatography coupled with mass spectrometry (UPLC–MS) that specifically detects all known DA ligands (e.g., Δ4-DA, Δ7-DA, and Δ1,7-DA). Surprisingly, comparing the UPLC–MS data to DA standards (*Table 1*) revealed the presence of Δ7-DA (*Figure 2B*) but no other congeners, such as Δ4-DA and Δ1,7-DA (*Figure 2—figure supplement 2*). The predicted endogenous concentration of Δ7-DA in *S. stercoralis* FL-L3 is ~200 nM, which is well above the concentration

**Table 1.** Detection methods for the compounds in this study.

| Steroids | Retention time (min) | MS detection mode | Parent ion (m/z) | Product ion (m/z) |
|---|---|---|---|---|
| Δ7-DA | 2.1 | Negative SIM | 413 | N/A |
| Δ4-DA | 1.9 | Negative SIM | 413 | N/A |
| Δ1,7-DA | 2.0 | Negative SIM | 411 | N/A |
| Δ7-DA-PA | 2.1 | Positive MRM | 505 | 487 |
| [$^{13}$C]-Δ7-DA-PA | 2.1 | Positive MRM | 508 | 490 |
| [$^2$H]–7-Dehydrocholesterol | 3.9 | Positive MRM | 374 | 109 |
| [$^2$H]-Lathosterone | 4.1 | Positive MRM | 392 | 109 |

MRM = multiple reaction monitoring. SIM = selective ion monitoring.

needed to fully activate Ss-DAF-12 (*Figure 2B*, inset). Taken together, these data demonstrate that Δ7-DA is present in *S. stercoralis* and acts as an endogenous DAF-12 ligand.

## Δ7-DA levels correlate with reproductive development in *S. stercoralis* lifecycle

In *C. elegans*, DAF-12 ligands promote reproductive development (*Bethke et al., 2009*; *Hammell et al., 2009*; *Motola et al., 2006*) and adulthood longevity (*Gerisch et al., 2007*; *Yamawaki et al., 2010*). To examine where in the *S. stercoralis* lifecycle Δ7-DA is present, lipid extracts from each developmental stage were analyzed by UPLC–MS (*Figure 3*). In the postparasitic environment under conditions that promote FL development, Δ7-DA was present in FL-L1/L2 and was significantly more abundant in the FL-L3 stage as the larvae progress toward reproductive maturity (*Figure 3A–C*). Similarly, Δ7-DA was detected in L3+ that have just infected the host and begun their reproductive development (*Figure 3A–C*). These findings are consistent with what is observed in *C. elegans* undergoing normal reproductive development (*Li et al., 2013*; *Motola et al., 2006*). A unique aspect of *S. stercoralis* is that after one FL generation outside of the host, PFL larvae are fated to growth arrest at L3i. In concordance with the hypothesis that developmental arrest is due to the absence of a DAF-12 ligand, Δ7-DA was undetectable in FL-adults, PFL-L1, and L3i larvae (*Figure 3A–C*).

The intestinal (Int)-L1-L3a larvae that were the direct progeny of parasitic adults in hyperinfected gerbils also had virtually no Δ7-DA (*Figure 3A–C*). These larvae that remain within the intestine are committed to become infectious as L3a, similar to their PFL counterparts that become L3i (*Lok, 2007*; *Viney and Lok, 2015*). By comparison, in the external postparasitic environment under temperature conditions that mimic the host intestine, the PP-L1 are also fated to become infectious as L3i (*Figure 3C*; *Albarqi et al., 2016*). The absence of Δ7-DA in these two stages (L3i and L3a) is similar to the developmental diapause observed in *C. elegans*, in which larvae arrest at the L3d stage in the absence of the DAF-12 ligand (*Motola et al., 2006*). Taken together with our previous work demonstrating that parasites lacking Ss-DAF-12 fail to respond to Δ7-DA and cannot enter or exit the L3i stage (*Cheong et al., 2021*), we conclude that liganded DAF-12 in *S. stercoralis* is required for reproductive development, while unliganded DAF-12 is requisite for the production of infectious L3i and L3a larvae.

Another intriguing finding was that the highest concentration of Δ7-DA occurred in parasitic adults, as opposed to the essentially undetectable levels in FL adults (*Figure 3B*). This difference in Δ7-DA levels correlated with the markedly different lifespans of the two adult populations. Parasitic adults live for up to a year or more in the host, whereas FL adults survive for only a few days in soil (*Lok, 2007*; *Viney and Lok, 2015*). Given that Δ7-DA is known to extend lifespan of adult *C. elegans* (*Gerisch et al., 2007*; *Yamawaki et al., 2010*), this finding suggests that Δ7-DA may play a similar role in extending the lifespan of parasitic adults in *S. stercoralis*.

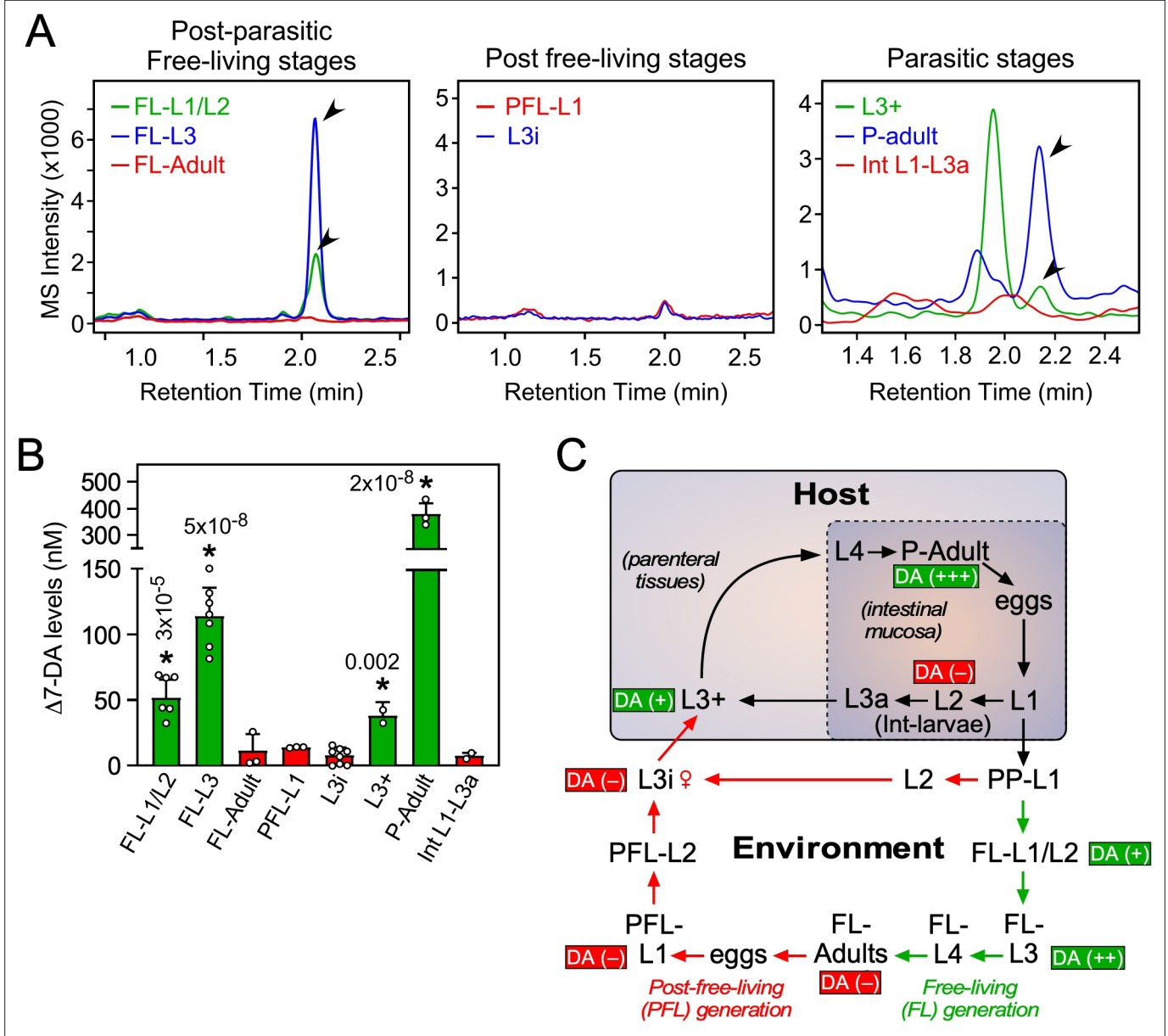

**Figure 3.** Profiling of Δ7-DA in developmental stages of *S. stercoralis*. (**A**) Detection of Δ7-DA during the lifecycle of *S. stercoralis*. Lipid extracts from the indicated stages of *S. stercoralis* were analyzed by derivatizing Δ7-DA to Δ7-DA-picolylamine, which then was detected by ultra-performance liquid chromatography coupled with mass spectrometry (UPLC–MS) in positive multiple reaction monitoring (MRM) mode with *m/z* transition 505 → 487. Parasitic stages were recovered from hyperinfected gerbils. Arrowheads show Δ7-DA peaks. Note that in the extracts from L3+ larvae, the peak with the faster retention time (1.95 min) is an unknown metabolite that is only found at this stage. (**B**) Δ7-DA levels were determined in the stages shown in (**A**) by comparison to a known standard. Data are presented as the mean ± standard deviation (SD) (*n* = 2–8); *p values (shown in the figure) were determined by Student's *t*-test compared to L3i larvae. (**C**) Schematic summary of Δ7-DA levels in the development stages of *S. stercoralis*. Δ7-DA is absent in larvae developing to infectious stages (i.e., L3i and L3a) and is present in larvae undergoing reproductive development.

## Identification of the Δ7-DA biosynthetic pathway in *S. stercoralis*

In *C. elegans*, DAF-12 ligands are synthesized from dietary cholesterol by a cascade of enzymes including a Rieske oxygenase (DAF-36), a short-chain hydroxysteroid dehydrogenase (DHS-16), and a cytochrome P450 (CYP; DAF-9) that catalyzes the final and rate-limiting step in dafachronic acid synthesis (*Figure 4A*; *Motola et al., 2006*; *Rottiers et al., 2006*; *Wollam et al., 2012*; *Wollam et al., 2011*). Bioinformatic analysis of the *S. stercoralis* genome revealed single homologs for both DAF-36 and DHS-16 (*Stoltzfus et al., 2012*), which share sequence identity with their *C. elegans* counterparts.

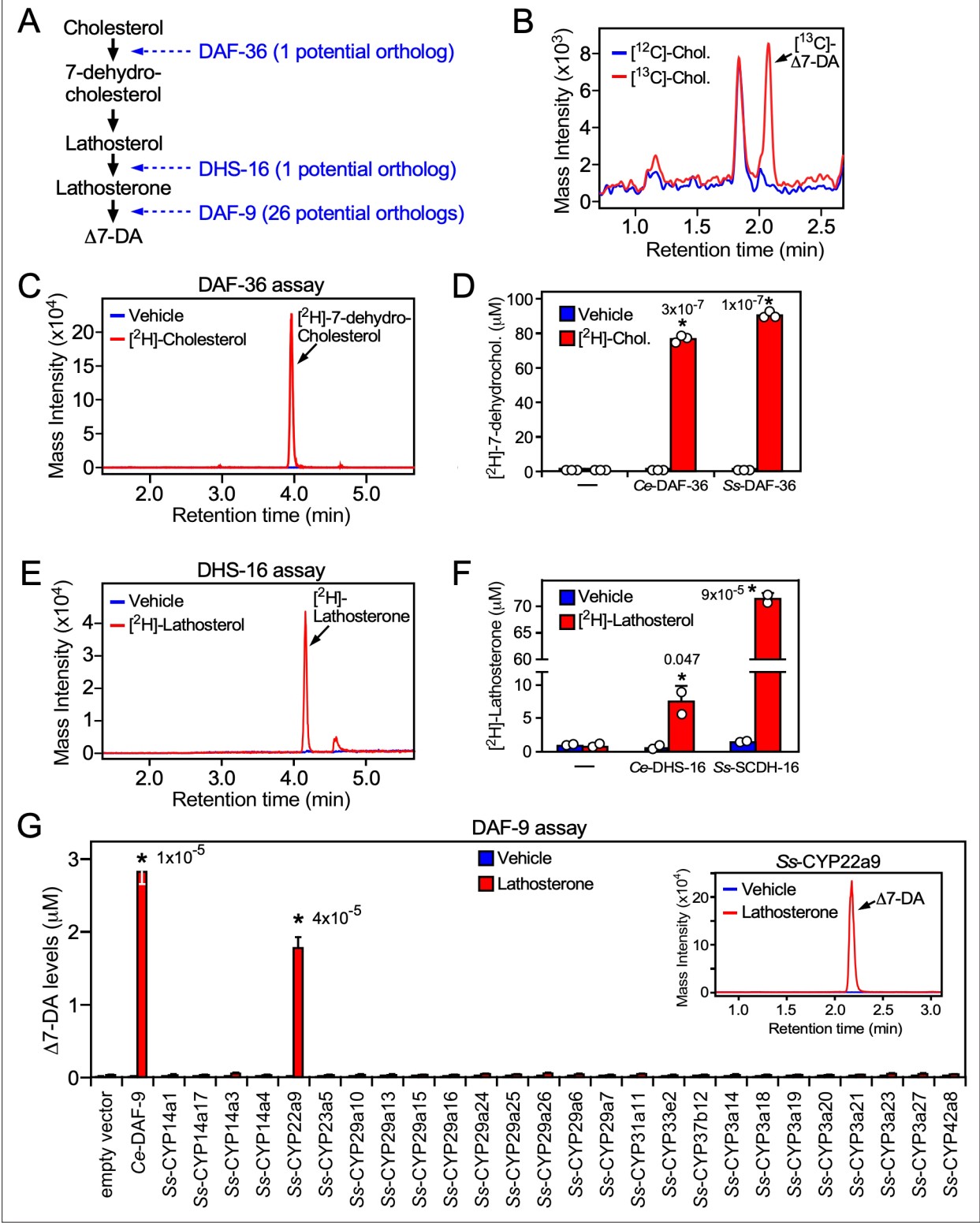

**Figure 4.** Characterization of the Δ7-DA biosynthetic pathway in *S. stercoralis*. (**A**) Diagram of Δ7-DA biosynthetic pathway in *C. elegans*. In blue are the known *C. elegans* enzymes followed in parentheses by the number of candidate orthologs found in *S. stercoralis*. (**B**) Δ7-DA is synthesized de novo in *S. stercoralis*. Extracts from FL-L3 worms cultured from PP-L1s in the presence or absence of [¹³C]-cholesterol were assayed by ultra-performance liquid chromatography coupled with mass spectrometry (UPLC–MS) for incorporation of [¹³C] into Δ7-DA. (**C, D**) *Ss*-DAF-36 catalyzes the synthesis of 7-dehydrocholesterol in the first step of Δ7-DA biosynthesis. Sf9 microsomes expressing *Ss*-DAF-36 were incubated with vehicle or 100 μM [²H]-cholesterol and assayed for the synthesis of [²H]-7-dehydrocholesterol by UPLC–MS chromatography (**C**), and the amount quantitated relative to that

*Figure 4 continued on next page*

*Figure 4 continued*

produced by the *C. elegans* ortholog, *Ce*-DAF-36 (**D**). (**E, F**) *Ss*-SCDH-16 catalyzes the of synthesis of lathosterone in the penultimate step of Δ7-DA biosynthesis. Sf9 microsomes expressing *Ss*-SCDH-16 were incubated with vehicle or 100 μM [$^2$H]-lathosterol and assayed for the synthesis of [$^2$H]-lathosterone by UPLC–MS chromatography (**E**), and the amount quantitated relative to that produced by the *C. elegans* ortholog, *Ce*-DHS-16 (**F**). (**G**) *Ss*-CYP22a9 catalyzes the synthesis of Δ7-DA from lathosterone. Sf9 cells expressing one of each of the 26 *S. stercoralis* cytochrome P450 homologs were incubated with vehicle or 10 μM lathosterone and assayed for the production of Δ7-DA by UPLC–MS as in *Figure 2*. *Ce*-DAF-9 is shown as a positive control. Inset, chromatogram from the reaction with *Ss*-CYP22a9. Data are presented as the mean ± standard deviation (SD) (*n* = 2–3); *p values (shown in the figure) were determined by Student's *t*-test compared to vehicle. See also *Figure 4—figure supplement 1*.

The online version of this article includes the following source data and figure supplement(s) for figure 4:

**Figure supplement 1.** Expression of the 26 *S. stercoralis* cytochrome P450 enzymes in insect Sf9 cells.

**Figure supplement 1—source data 1.** Full gel images for the expression of *Ss-CYPs*.

In contrast, there are 26 CYPs in *S. stercoralis*, but none of these share predictive sequence identity with DAF-9 (~36% at the best) (*Stoltzfus et al., 2012*). The lack of an obvious DAF-9 ortholog suggested the possibility that Δ7-DA might be acquired exogenously by *S. stercoralis*. To rule out this possibility, we first determined whether *S. stercoralis* synthesize Δ7-DA from dietary cholesterol in vivo. FL PP-L1 were cultured on a diet where the natural cholesterol isotope (largely $^{12}$C-cholesterol) was supplemented with $^{13}$C-labeled cholesterol. After 24 hr, we detected significant levels of $^{13}$C-labeled Δ7-DA in the FL-L3 (*Figure 4B*). These results demonstrated that Δ7-DA is generated from dietary cholesterol in *S. stercoralis* and that the parasite has all of the necessary enzymatic machinery to perform the synthesis.

To characterize the biosynthetic enzymes, we employed a Sf9 cell expression system that we used to identify the *C. elegans* counterparts of these proteins (*Motola et al., 2006*; *Rottiers et al., 2006*; *Wollam et al., 2012*; *Wollam et al., 2011*). To avoid potential interference from any endogenous substrates in Sf9 cells, we utilized $^2$H-isotope-labeled substrates to assay the activity of the candidate *S. stercoralis* orthologs, *Ss*-DAF-36 (SSTP_0000037900) and *Ss*-SCDH-16 (SSTP_0001031100, the homolog of *Ce*-DHS-16). Sf9 microsomes expressing *Ss*-DAF-36 readily converted cholesterol to 7-dehydrocholesterol (*Figure 4C*) with an efficacy comparable to the *C. elegans* enzyme (*Figure 4D*). Similarly, *Ss*-SCDH-16 catalyzed the conversion of lathosterol to lathosterone like *Ce*-DHS-16 (*Figure 4E, F*). These findings demonstrated that *S. stercoralis* has functional orthologs of DAF-36 and DHS-16.

The inability to predict the parasite's DAF-9 ortholog from our bioinformatic analysis prompted us to perform an unbiased screen to identify the enzyme from among all 26 *S. stercoralis* CYPs. Each of the 26 CYP enzymes was expressed in Sf9 cells (*Figure 4—figure supplement 1*) and assayed by UPLC–MS for their ability to synthesize Δ7-DA from lathosterone. Unambiguously, only one of the 26 CYPs, *Ss*-CYP22a9 (SSTP_0001032100) synthesized Δ7-DA at levels comparable to *Ce*-DAF-9 (*Figure 4G*). These results demonstrated that *Ss*-CYP22a9 is the DAF-9 isoenzyme in *S. stercoralis*.

## Δ7-DA synthesis is required for reproductive development in *S. stercoralis*

In *C. elegans*, favorable environmental cues induce DAF-9 expression and DAF-12 ligand synthesis through a cGMP signaling pathway, leading to reproductive development (*Hu, 2007*). Likewise in *S. stercoralis*, favorable host conditions act through a similar pathway to stimulate feeding, a sign of reactivated development in L3i (*Stoltzfus et al., 2014*). To study the role of Δ7-DA synthesis in parasite reproductive development, we first asked whether Δ7-DA is made in response to activation of the cGMP pathway in L3i. To that end, we treated L3i with 8-Br-cGMP, which mimics the presence of the host environment by activating the cGMP signaling pathway, and then tested endogenous Δ7-DA levels in the L3i larvae. In the presence of 8-Br-cGMP, we found that endogenous Δ7-DA was increased progressively over a 3-day period to levels (~100 nM) that would saturate *Ss*-DAF-12 occupancy (*Figure 5*). Furthermore, this stimulation of Δ7-DA synthesis correlated exactly with the expression of *Ss-cyp22a9* (*Figure 5—figure supplement 1A*) and was abolished by cotreatment with ketoconazole, a broad-spectrum CYP inhibitor (*Figure 5*). These results highlight the importance of Δ7-DA synthesis in the reactivation of L3i larval development.

We next tested the requirement of *Ss*-CYP22a9 for Δ7-DA synthesis and parasite development in vivo. For these experiments, a CRISPR-mediated, homology-directed gene knockout strategy was

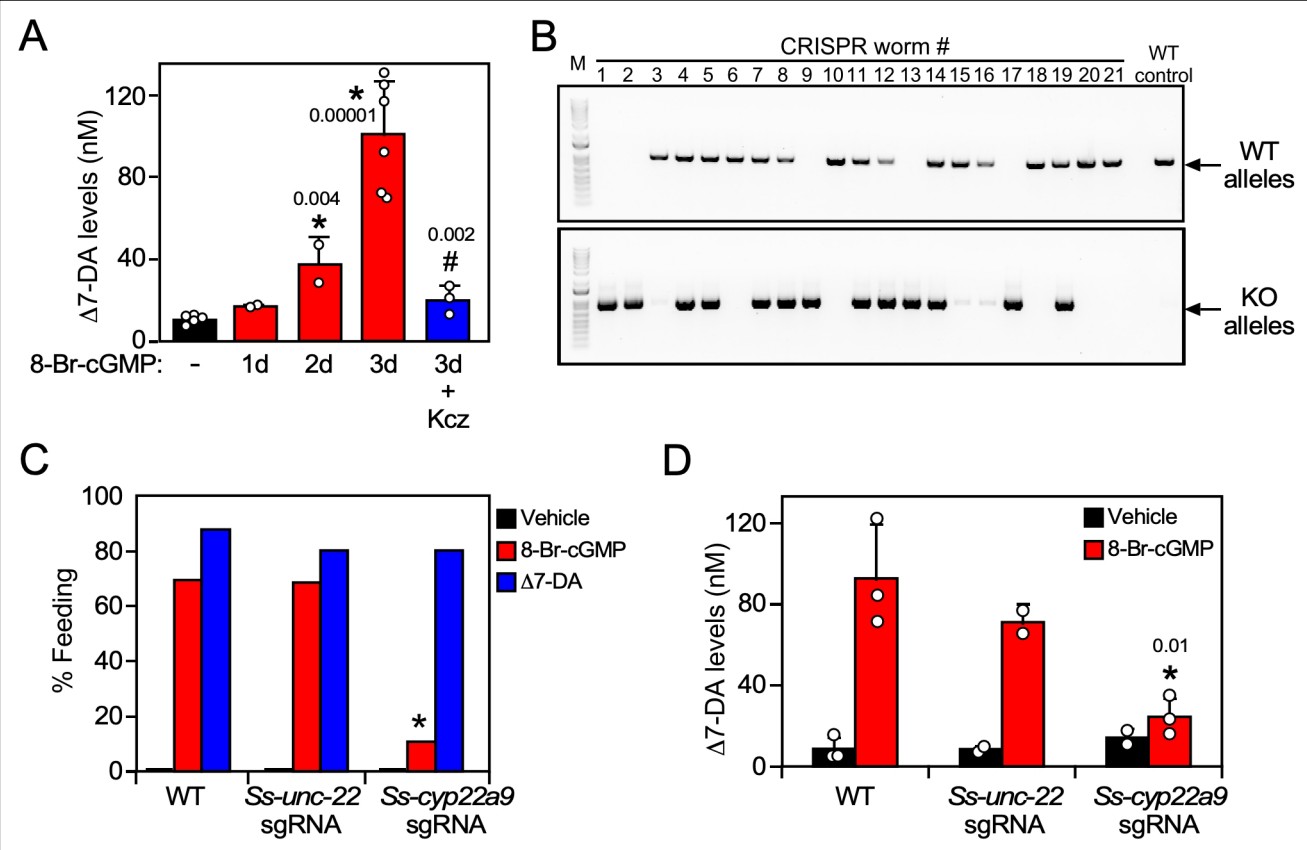

**Figure 5.** *Ss*-CYP22a9 is required for L3i activation and Δ7-DA synthesis in *S. stercoralis*. (**A**) Inhibition of cytochrome P450 activity blocks Δ7-DA synthesis in parasites. L3i (1000 worms/group) were treated with 0.5 mM 8-Br-cGMP in the presence or absence of 25 μM ketoconazole (Kcz). Data are presented as the mean ± standard deviation (SD) (*n* = 2–6); p values (shown in the figure) were determined by *t*-test compared to vehicle (*) or 3-day treatment with 8-Br-cGMP (#). (**B, C**) *Ss*-CYP22a9 is required for cGMP-induced L3i activation. The *Ss-cyp22a9* gene was disrupted by CRISPR/Cas9-mediated, homology-directed repair as shown in *Figure 5—figure supplement 1C*. The resulting F1 generation of L3i worms expressing the positive selection marker (GFP) were sorted manually, subjected to single worm genotyping (**B**) and assayed for feeding behavior (**C**). Disruption of the *Ss-unc-22* gene was used as a control. *p = 1 × 10⁻¹² compared to wild-type (WT) and p = 1 × 10⁻⁷ compared to *Ss-unc-22* by Fisher's exact test (*n* = 40–200 from three independent experiments). For full gel blot images, see *Figure 5—source data 1*. (**D**) Δ7-DA synthesis is abolished in *Ss-cyp22a9* knockout parasites. *Ss-cyp22a9* or *Ss-unc-22* (as a control) genes were disrupted by CRISPR using the same sgRNA plasmids in (**B, C**) and F1 generation L3i worms were assayed for Δ7-DA levels as in *Figure 3* after treatment with 0.5 mM 8-Br-cGMP. Data represent the mean ± SD (*n* = 2–3); *p < 0.03 by *t*-test compared to WT or *Ss-unc-22* worms treated with 8-Br-cGMP. See also *Figure 5—figure supplement 1*.

The online version of this article includes the following figure supplement(s) for figure 5:

**Source data 1.** Full gel images for single worm genotyping.

**Figure supplement 1.** In vivo characterization of the DAF-9 homolog (*Ss*-CYP22a9) in *S. stercoralis*.

used to insert a GFP expression cassette into the *Ss-cyp22a9* gene locus through homologous repair (*Figure 5—figure supplement 1B, C*). This strategy allowed us to enrich for *Ss-cyp22a9* knockout worms by selecting GFP-positive L3i. Single worm genotyping confirmed the presence of at least one disrupted *Ss-cyp22a9* allele in >60% of the GPF-positive L3i (*Figure 5B*). To evaluate the phenotype due to loss of *Ss*-CYP22a9, we assayed cGMP-induced feeding behavior in L3i, which as noted above is an early marker of the transition to reproductive development (*Lok, 2007*; *Stoltzfus et al., 2014*; *Viney and Lok, 2015*). In both wild-type larvae and CRISPR-control larvae in which a gene unrelated to development (*Ss_unc-22*) was targeted, 8-Br-cGMP strongly induced the feeding behavior as expected (*Figure 5C*). In contrast, in *Ss-cyp22a9* knockout larvae, cGMP-induced feeding behavior was severely impaired (*Figure 5C*). The residual cGMP-induced response is likely due to the presence of an incomplete penetrance of the knockout in all worms. The loss of this feeding behavior in *Ss-cyp22a9*-deficient larvae was rescued completely by treatment with exogenous Δ7-DA, further demonstrating the importance of DAF-12 activation as a requirement for parasite development

(*Figure 5C*). To confirm the enzymatic function of *Ss*-CYP22a9, we also analyzed the production of Δ7-DA in *Ss-cyp22a9* knockout L3i larvae. For this experiment, we targeted the *Ss-cyp22a9* gene using nonhomologous end joining CRISPR with the same sgRNA as in *Figure 5C* and *Figure 5—figure supplement 1C*. This CRISPR method permitted the generation of larger numbers of *Ss-cyp22a9* knockouts, which were necessary to detect endogenous Δ7-DA levels. Consistent with the results of the CYP inhibitor, disrupting the *Ss-cyp22a9* gene abolished the Δ7-DA induction by cGMP (*Figure 5D*). Taken together, these results demonstrated the crucial role of *Ss*-CYP22a9 in Δ7-DA synthesis and highlight the importance of Δ7-DA synthesis in the development of infectious larvae in *S. stercoralis*.

## Δ7-DA suppresses parasitic burden in uncomplicated strongyloidiasis

In *S. stercoralis*, uncomplicated infections are frequently asymptomatic but can last for years due to the persistence of L3a worms that escape the host immune system and continually reinfect the host at a low level. However, when the host is immune suppressed and the intestinal barrier is compromised, autoinfection accelerates out of control as more and more L3a worms invade the host, leading to a lethal hyperinfection. The results above suggested that targeting the DAF-12 pathway might provide a novel therapeutic opportunity for treating this disease. This strategy is based on the idea that like their L3i counterparts in the environment, the lack of Δ7-DA is required for intestinal larvae to become infectious L3a (*Figure 3C*). Thus, we postulated that pharmacological activation of DAF-12 in intestinal larvae should block autoinfection by preventing the formation of filariform L3a larvae as it does with L3i in the environment (*Albarqi et al., 2016*; *Wang et al., 2009*). We first asked whether Δ7-DA can successfully treat the uncomplicated, latent form of the disease. To that end, we employed a well-established gerbil model that mimics both uncomplicated and hyperinfective strongyloidiasis found in humans (*Kerlin et al., 1995*; *Nolan et al., 1993*).

For the uncomplicated model, we infected gerbils with 1000 L3i and after 21 days began administering vehicle or Δ7-DA orally in their drinking water, which delivers the compound directly into the gut where the intestinal larvae reside. We then monitored the parasite burden of the intestine by counting the number of fecal larvae. Compared to vehicle, treatment with Δ7-DA for 14 days dramatically reduced fecal larval output by ~90% (312 ± 103 larvae per gram feces for vehicle vs 32 ± 6 for DA, *q* < 0.03; *Figure 6*). In contrast, the numbers of parasitic adults, which produce intestinal larvae, were not significantly changed by Δ7-DA treatment (*Figure 6B*). Because of the time-frame of this experiment and the fact that autoinfection is minimal in the uncomplicated model (*Nolan et al., 1993*), reinfection of the host by L3a larvae is negligible, demonstrating that these adult parasites

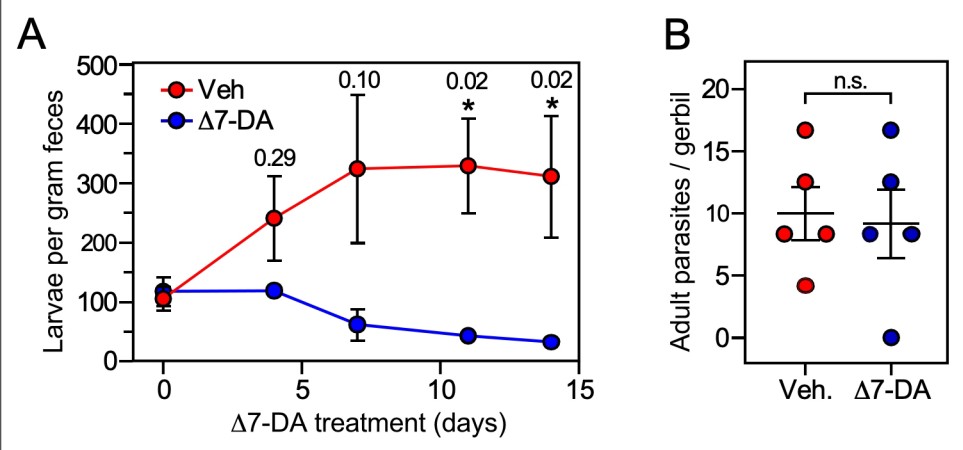

**Figure 6.** Δ7-DA suppresses output of fecal larvae in latent, uncomplicated strongyloidiasis. (**A**) Δ7-DA treatment reduces fecal larvae by >90% in gerbils infected with *S. stercoralis*. (**B**) Adult parasite burden in infected gerbils measured at 14-day post-treatment. Data are plotted as the mean ± standard error (SE) (*n* = 5); *q* values (shown in the figure) were determined by Mann–Whitney *U* test compared to vehicle (*statistically significant; n.s., not significant). See also *Figure 6—figure supplement 1* for individual data points in (**A**).

The online version of this article includes the following figure supplement(s) for figure 6:

**Figure supplement 1.** Δ7-DA suppresses output of fecal larvae in latent, uncomplicated strongyloidiasis.

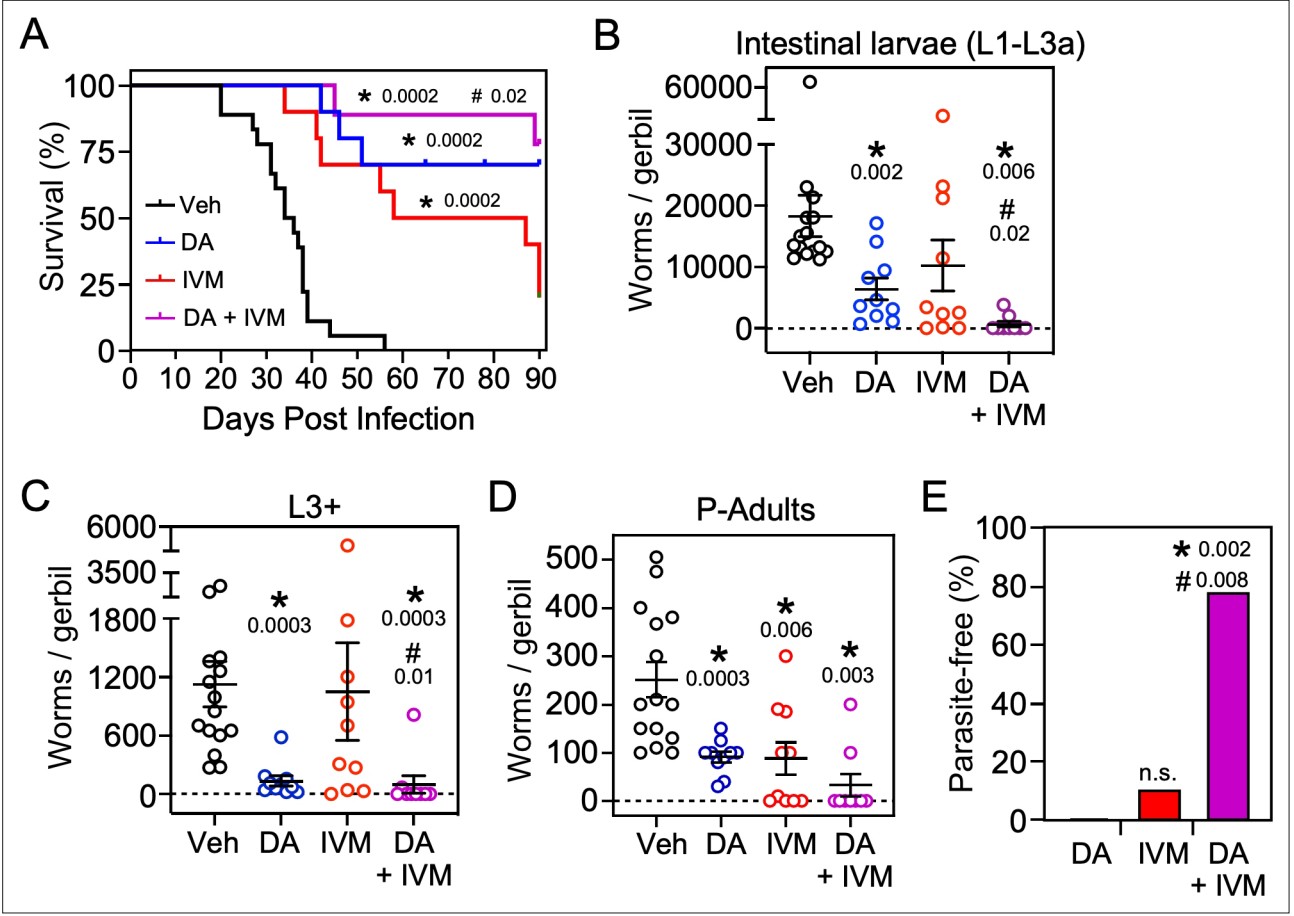

**Figure 7.** Δ7-DA and ivermectin act cooperatively to treat disseminated strongyloidiasis hyperinfection. (**A**) Kaplan–Meier survival curves of hyperinfected gerbils treated with vehicle (Veh), Δ7-DA, and/or ivermectin (IVM). Sample sizes: n = 18 (Veh), 10 (DA), 10 (IVM), 9 (DA+ IVM); q values (shown in the figure) were determined by log-ranked test compared to treatment with vehicle (*) or ivermectin alone (#). (**B–D**) Parasite burden at various lifecycle stages from the hyperinfected gerbil treatment groups shown in (**A**). At the time of death, live intestinal L1-L3a (**B**), L3+ (**C**), and adult (**D**) parasites were counted. Sample sizes: n = 15 (Veh), 10 (DA), 10 (IVM), 9 (DA+ IVM); q values (shown in the figure) were determined by Mann–Whitney U test compared to treatment with vehicle (*) or ivermectin alone (#). (**E**) Cotreatment with Δ7-DA and ivermectin eradicates parasites in hyperinfected gerbils. Animals from (**A**) with no detectable parasites in any part of the body after 90 days of treatment were scored as parasite-free. Notably, all the animals in the cotreatment group that survived hyperinfection (shown in (**A**)) were found to be parasite-free. Sample sizes: n = 10 (DA), 10 (IVM), 9 (DA+ IVM); q values (shown in the figure) were determined by Fisher's exact test compared to treatment with vehicle (*) or ivermectin alone (#); n.s., not significant. DA, 50 µM Δ7-DA administered in drinking water; IVM, 300 µg/kg ivermectin by i.p. injection. See also ***Figure 7—figure supplement 1***.

The online version of this article includes the following figure supplement(s) for figure 7:

**Figure supplement 1.** Ivermectin (IVM) mimics the treatment of a human *S. stercoralis* hyperinfection in gerbil models.

were derived from the L3i of the initial infection. This explains why the numbers of adults (in contrast to the intestinal larvae) were not substantially reduced by Δ7-DA treatment. These results are also consistent with our observation that intestinal larvae lack Δ7-DA (***Figure 3***), which is essential for L3a development, thereby supporting the hypothesis that Δ7-DA specifically targets the intestinal larval stages of the lifecycle. We conclude that Δ7-DA treatment impedes the development of intestinal larvae and thus blocks transmission in uncomplicated infections where conventional treatments are only partially effective.

## Δ7-DA improves survival of a lethal strongyloidiasis hyperinfection

Based on the finding that Δ7-DA disrupts intestinal larval development (***Figure 6A***), we next asked whether the DAF-12 ligand would be effective at treating the lethal form of strongyloidiasis. For this experiment, beginning at the time of infection with L3i, gerbils were administered the glucocorticoid, methylprednisolone, on a weekly basis (***Nolan et al., 1993***). Similar to what is observed

in immunocompromised humans, this results in the rapid progression of disease to a disseminated hyperinfection that is lethal to all of the animals, with >90% of them dying within 40 days (*Figure 7*, black line). In contrast, when a matched group of these animals was treated with Δ7-DA starting at 10 days after the hyperinfection was initiated, there was an impressive 70% survival rate that lasted for at least 90 days at which point we arbitrarily terminated the experiment (*Figure 7A*, blue line). Autopsies of these animals showed that hyperinfection resulted in a massive increase in the number of viable intestinal parasites (>10$^4$) in vehicle-treated gerbils, which was reduced substantially in Δ7-DA-treated animals (*Figure 7A-D*). Notably, the number of autoinfected L3+ larvae in parenteral tissues, which drive the resulting pathogenesis, was reduced by 90% (and in some cases was undetectable) in Δ7-DA-treated animals compared to vehicle-treated animals (*Figure 7C*). These results are consistent with what was observed in the uncomplicated infection and supports the conclusion that Δ7-DA treatment impedes the development of L3a. Compared to vehicle-treated gerbils with uncomplicated infections (*Figure 6B*), we observed a >10-fold increase in adult parasites in hyperinfected vehicle-treated gerbils (*Figure 7D*). However, unlike in the uncomplicated infection, there was a significant decrease in adult parasites in the Δ7-DA-treated gerbils that had a hyperinfection (50% vs. the vehicle treated, *Figure 7D*). This difference is because in a hyperinfection the parasitic adults are largely derived from the expanding L3a population, whereas in an uncomplicated infection the number of adults that were present matured directly from the L3i larvae used for the initial infection. Overall, these findings demonstrated that Δ7-DA treatment is effective in treating a lethal *S. stercoralis* hyperinfection.

## Δ7-DA and ivermectin act cooperatively to treat strongyloidiasis hyperinfection

As the current front-line drug for treating strongyloidiasis, ivermectin is effective at removing adult *S. stercoralis* parasites (*Krolewiecki et al., 2013*; *Repetto et al., 2018*). However, in hyperinfected patients ivermectin treatment is only 50–60% effective at preventing death (*Buonfrate et al., 2013*).

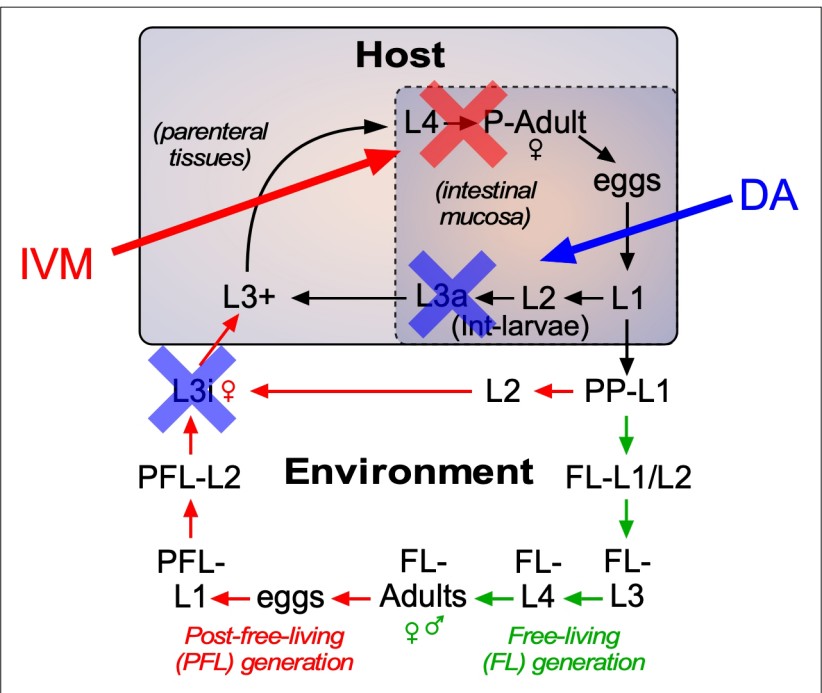

**Figure 8.** Strategy for using DAF-12-based therapeutics to treat strongyloidiasis. Administration of DAF-12 ligands like Δ7-DA disrupts the lifecycle of nematode parasites by preventing the development of the infective L3i and L3a worms, where DAF-12 is normally unliganded. Pharmacologic activation of DAF-12 with Δ7-DA prevents both environmental infection and autoinfection, which are essential features of the latent and hyperinfection forms of strongyloidiasis. In contrast, ivermectin kills only the actively developing stages and therefore is unable to target developmentally-quiescent infective L3i and L3a larvae. For this reason, the combination of the DAF-12 ligand and ivermectin achieves a synergistic, double blockade of the lifecycle.

This high mortality is due in part to the relatively low efficacy of ivermectin at killing the persistent autoinfective L3a larvae. Given that Δ7-DA specifically targets the L3a population, we hypothesized that the combination of Δ7-DA and ivermectin would act cooperatively to treat a strongyloidiasis hyperinfection. To test this, we first established an ivermectin treatment model in hyperinfected gerbils that mimics the efficacy observed in humans (*Figure 7—figure supplement 1*). In this model, 300 μg/kg ivermectin delivered i.p. acutely cleared fecal larva output in ~40% of the hyperinfected gerbils within 7 days, resulting in ~25% survival rate for at least 90 days (*Figure 7A*, red line). As expected, ivermectin was as effective as Δ7-DA in removing adult parasites from animals undergoing hyperinfection (*Figure 7D*), but in contrast to Δ7-DA, ivermectin was ineffective at decreasing either the intestinal or L3+ larval populations (*Figure 7B, C*). Next, we treated hyperinfected gerbils with both Δ7-DA and ivermectin. We found that the combination of both agents improved the survival rate significantly over either agent alone (>75%, *Figure 7A*) and reduced the parasite burden of all life stages (*Figure 7B–D*). Remarkably, the combination of Δ7-DA and ivermectin completely eliminated the parasites from all the animals that survived, essentially curing them of the disease (*Figure 7E*). In contrast, when treated with ivermectin or Δ7-DA alone, ≤10% were parasite-free even among those animals that survived (*Figure 7E*). Taken together, these results demonstrated that Δ7-DA and ivermectin act synergistically in successfully eliminating *S. stercoralis* by targeting complementary stages of the lifecycle (*Figure 8*).

## Discussion

The worldwide prevalence of *S. stercoralis* infections and the increased risk of deadly hyperinfection due to the broad use of glucocorticoids to treat inflammatory diseases underscores the urgent need for new therapeutic approaches to treat strongyloidiasis (*Buonfrate et al., 2013*; *Repetto et al., 2018*). The difficulty in treating the disease arises from a persistent autoinfection in which L3a are produced and continuously reinfect the host. Consequently, the parasite burden can become uncontrollably large when the immune system is compromised, leading to frequent lethality. Our previous studies have suggested one potential target of interest is the nuclear receptor, DAF-12 (*Patton et al., 2018*; *Wang et al., 2017*; *Wang et al., 2009*), which is required for the transition into and out of the L3i stage of the parasite's lifecycle (*Cheong et al., 2021*). Here, we addressed two key unanswered questions: what is the nature and role of the *Ss*-DAF-12 ligand, and is targeting the *Ss*-DAF-12 pathway a viable therapeutic strategy? In this report, we identified Δ7-DA as the endogenous ligand of *Ss*-DAF-12 and showed that the regulation of its biosynthesis is a requirement for both reproductive growth and the development of infectious stage larvae. Similar to the free-living nematode, *C. elegans*, we found the presence of Δ7-DA is a requisite feature for *S. stercoralis* FL and parasitic larvae to mature into reproductive adults. In contrast, we discovered Δ7-DA is absent during the formation of L3a. Thus, like the analogous development of L3i in the environment, unliganded DAF-12 appears to be required for L3a formation in hosts. Taken together, these findings revealed a vulnerability in the lifecycle of *S. stercoralis* and suggested two new therapeutic strategies: either blocking the synthesis of the ligand to prevent reproductive development or precociously activating DAF-12 in the intestine to prevent autoinfection.

To that end, we showed that knocking out the Δ7-DA biosynthetic enzyme prevents the initiation of reproductive development. Concomitantly, when gerbils infected with *S. stercoralis* were treated with Δ7-DA, the development of L3a was severely impaired, thereby suppressing the parasite's autoinfection cycle in both uncomplicated and hyperinfection forms of strongyloidiasis. Strikingly, when Δ7-DA was combined with the front-line medication, ivermectin, there was a near complete remission of the hyperinfection that resulted in a substantial increase in survival (from 25% with ivermectin alone to 75% with the combination treatment). Notably, in all of the surviving animals receiving both Δ7-DA and ivermectin, no parasites from any stage were detected anywhere in the body, suggesting that the cotreatment resulted in a complete cure.

### Advantages of DAF-12 as a therapeutic target

The synergistic response of Δ7-DA and ivermectin highlights the unique potential of targeting the DAF-12 signaling pathway and overcoming the limitations of using ivermectin alone. Mortality rates in ivermectin-treated patients with hyperinfection range as high as 50% (*Buonfrate et al., 2013*). As

an inhibitor of muscular function, ivermectin is highly effective at killing adults and feeding larvae (*Repetto et al., 2018*), but less effective at killing the developmentally quiescent L3a larvae that continuously drive autoinfection. In contrast, Δ7-DA directly interferes with infectious L3a larval development by binding to unliganded DAF-12 and provoking an inappropriate developmental response inside the host that results in death of the larvae. In this way, the two drugs complement each other by targeting different stages of the lifecycle (*Figure 8*). Targeting the ligand binding properties of DAF-12 has other distinct advantages in that this nuclear receptor is nematode specific and mutants resistant to pharmacologic ligands are unlikely to develop, since such mutants would be expected to disrupt the endogenous function of the receptor and otherwise be lethal.

## The conundrum of Δ7-DA as the *Ss*-DAF-12 ligand

An intriguing aspect of this study was the finding that Δ7-DA is the endogenous ligand for *Ss*-DAF-12. Although Δ7-DA was first identified as the endogenous DAF-12 ligand in *C. elegans*, its presence in *S. stercoralis* was unexpected. In comparison to its *C. elegans* homolog, the *Ss*-DAF-12 ligand binding domain shares only moderate sequence identity (42%) and exhibits a distinct pharmacologic profile in response to Δ7-DA compared to other species (*Wang et al., 2009*). The sequences of other regulatory proteins in the *Ss*-DAF-12 pathway also have diverged. For example, the cytochrome P450 (*Ss*-CYP22a9) that catalyzes the synthesis of Δ7-DA (*Figure 4*) and the ligand-dependent coactivator that *Ss*-DAF-12 requires for transactivation (*Ss*-DIP-1) (*Cheong et al., 2021*) are both unique to *Strongyloides* spp.

In spite of these differences, several lines of evidence indicate that Δ7-DA is the major, if not exclusive, DAF-12 ligand in *S. stercoralis*. First, no other ligand activities, including other types of dafachronic acids were detected. Second, the endogenous concentration of Δ7-DA in *S. stercoralis* (~200 nM) is sufficient to drive full activation of *Ss*-DAF-12, leaving other potential DAF-12 ligands redundant if they do exist. Finally, knocking out the *daf-9* homolog, *Ss-cyp22a9*, which is required to synthesize Δ7-DA, prevents DAF-12-dependent activity in vivo, and that activity is rescued completely by adding back the ligand. Interestingly, Δ7-DA has been shown to activate DAF-12 homologs in other species (*Ayoade et al., 2020*; *Long et al., 2020*; *Ma et al., 2019*; *Ogawa et al., 2009*; *Wang et al., 2009*). Why the Δ7-DA hormone has been evolutionarily conserved, whereas other components of the pathway have not, remains unclear, but nevertheless highlights the importance of the ligand in governing nematode biology.

## Therapeutic potential beyond *S. stercoralis*

The findings discussed above underscore the potential of developing agonists that would precociously activate unliganded *Ss*-DAF-12 and kill L3a larvae, a therapy that would be propitious and specific to *S. stercoralis*. However, the observation that liganded DAF-12 also is required at a completely different but essential stage of the lifecycle to promote reproductive growth raises the possibility of also targeting DAF-12 with an antagonist. Given that DAF-12 has been found in all nematode species surveyed to date (*Ayoade et al., 2020*; *Long et al., 2020*; *Ma et al., 2019*; *Ogawa et al., 2009*; *Wang et al., 2009*), antagonizing the receptor would be expected to prevent development of adult worms in all parasitic species, offering a unique, broad-spectrum approach for treating most, if not all, nematode parasitisms.

In addition to targeting the receptor, the identification of *Ss*-CYP22a9 as the Δ7-DA-synthesizing enzyme that is required for reproductive development reveals the potential of developing an enzyme inhibitor. Such an inhibitor would prevent synthesis of the endogenous ligand and thus the requisite activation of DAF-12 for reproductive growth. Indeed, the finding that the broad-spectrum CYP-inhibitor, ketoconazole, was effective at blocking Δ7-DA synthesis in the present study and was shown previously to block developmental activation of cultured L3i (*Albarqi et al., 2016*) supports this strategy. The strategy for developing CYP inhibitors as drugs is well established (*Francis and Delgoda, 2014*).

While our study provides a proof of concept for the therapeutic use of an *Ss*-DAF-12 ligand, we note that Δ7-DA may not be the best candidate for eventual clinical use. Typical of most endogenous steroid receptor ligands, Δ7-DA has a short half-life and relatively poor pharmacokinetic properties in vivo (*Patton et al., 2018*). For this reason, in our study we administered Δ7-DA continuously in the drinking water over many days. Although overcoming this limitation would be important for

developing a suitable therapeutic modality, this is not an unrealistic goal. Encouragingly, in the case of all other nuclear receptors this limitation has been circumvented through the design of potent, long-lasting receptor agonists and antagonists (*Chen, 2008*; *Zhao et al., 2019*). Another limitation of this study is that we only tested single doses of Δ7-DA and ivermectin. However, we note that optimizing the dose regimens are likely to reveal even greater cooperative effects. Finally, we have not definitively ruled out the existence of other endogenous DAF-12 ligands. While the findings that Δ7-DA is the only ligand we detected and is present only at the times when DAF-12 would be expected to be an activator are consistent with the conclusion that Δ7-DA is the relevant ligand, it is feasible that other activities exist that were not identified in our purification scheme.

## Conclusions

In summary, in this work we identified the endogenous DAF-12 ligand in the human parasitic nematode *S. stercoralis*, characterized the ligand's biosynthesis pathway, and demonstrated that activating DAF-12 can overcome the limitation of the front-line drug, ivermectin, and significantly improve the outcome of strongyloidiasis in a preclinical model of hyperinfection. Our study further revealed the ligand's biosynthetic enzyme as a potential new target that might also be exploited to treat this lethal parasitic disease.

# Materials and methods

**Key resources table**

| Reagent type (species) or resource | Designation | Source or reference | Identifiers | Additional information |
|---|---|---|---|---|
| Gene (*S. stercoralis*) | *Ss_cyp22a9* | Wormbase | SSTP_0001032100 | |
| Gene (*S. stercoralis*) | *Ss-daf-36* | Wormbase | SSTP_0000037900 | |
| Gene (*S. stercoralis*) | *Ss-scdh-16* | Wormbase | SSTP_0001031100 | |
| Cell line (*Cercopithecus aethiops*) | COS-7 | ATCC | Cat# CRL-1651; RRID:CVCL_0224 | |
| Cell line (*Spodoptera frugiperda*) | Sf9 | ATCC | Cat# CRL-1711; RRID:CVCL_0549 | |
| Strain, strain background (*Meriones unguiculatus*, male) | Mongolian gerbil | Charles Rivers | Strain code: 243 | |
| Strain, strain background (*Canis familiaris*, male) | Dog | Oak Hill Genetics | N/A | |
| Antibody | anti-HA tag antibody [HA.C5] (Mouse monoclonal) | Abcam | Cat# ab18181; RRID:AB_444303 | WB (1:2000) |
| Recombinant DNA reagent | pGL4.53 (plasmid) | Promega | E5011 | |
| Recombinant DNA reagent | pCMX-CeSs-DAF-12 (plasmid) | *Wang et al., 2009* (PMID:19497877) | N/A | |
| Recombinant DNA reagent | pNL3.1-DAF-12RE-lit-1 (plasmid) | This paper | N/A | DAF-12 reporter |
| Recombinant DNA reagent | pFastBac-Dual-hOR (plasmid) | *Motola et al., 2006* (PMID:16529801) | N/A | |

*Continued on next page*

*Continued*

| Reagent type (species) or resource | Designation | Source or reference | Identifiers | Additional information |
|---|---|---|---|---|
| Recombinant DNA reagent | pFastBac-Dual-hOR-CYPs (plasmids) | This paper | N/A | Express hOR and CYPs in Sf9 cells |
| Recombinant DNA reagent | pFastBac-Dual-hOR-DAF-36s (plasmids) | This paper | N/A | Express hOR and DAF-36s in Sf9 cells |
| Recombinant DNA reagent | pFastBac-Dual-hOR-DHS-16s (plasmids) | This paper | N/A | Express hOR and DHS-16s in Sf9 cells |
| Recombinant DNA reagent | pML60-*Ss-unc-22* (plasmid) | **Gang et al., 2017** (PMID:29016680) | N/A | |
| Recombinant DNA reagent | pML60-*Ss-cyp22a9* (plasmid) | This paper | N/A | Guide RNA plasmid for *Ss-cyp22a9* |
| Recombinant DNA reagent | pPV540 (plasmid) | **Lok, 2019** (PMID:31379923) | N/A | |
| Commercial assay or kit | Nano-Glo Dual-luciferase kits | Promega | N1620 | |
| Commercial assay or kit | NADPH regeneration system | Promega | V9510 | |
| Commercial assay or kit | Microsome Isolation Kit | Abcam | ab206995 | |
| Chemical compound, drug | Ketoconazole | Sigma | K1003 | |
| Chemical compound, drug | 8-Bromo-cGMP | Tocris | 1089 | |
| Chemical compound, drug | Methylprednisolone acetate | Zeotis | DEPO-MEDRO | 20 mg/ml |
| Chemical compound, drug | Ivermectin | Merial Limited | Ivermec | 1% solution |
| Chemical compound, drug | Chenodeoxycholic acid-$^2H_4$ | Sigma | 614,122 | |
| Chemical compound, drug | Triphenylphosphine | Sigma | T84409 | |
| Chemical compound, drug | 2,2'-Dipyridyl disulfide | Sigma | D5767 | |
| Chemical compound, drug | 2-Picolylamine | Sigma | A65204 | |
| Chemical compound, drug | Cholesterol-$^{13}C_3$ | Cambridge Isotope | CLM-9139 | |
| Chemical compound, drug | Cholesterol-$^2H_7$ | Avanti Polar Lipids | 700041 P | |
| Chemical compound, drug | Lathosterol-$^2H^7$ | Avanti Polar Lipids | 700,056 | |
| Chemical compound, drug | 7-Dehydrocholesterol-$^2H_7$ | Avanti Polar Lipids | 700116P | |
| Chemical compound, drug | Lathosterone | Steraloids | C7500-000 | |
| Chemical compound, drug | Alexa Fluor 594 fluorescent dye | Thermo Fisher | A33082 | |

## Animal husbandry

All animal experiments were approved by the University of Texas Southwestern Medical Center Institutional Animal Care and Use Committee (IACUC) and are listed under animal protocol numbers 2016-101500 and 2018-102369. All protocols, as well as routine husbandry care of the animals, were conducted in strict accordance with the Guide for the Care and Use of Laboratory Animals of the National Institutes of Health (*NationalResearchCouncil, 2011*). Male dogs of 6–18 months were purchased from Oak Hill Genetics (Ewing, IL) and infected with 3000 L3i of *S. stercoralis* and were orally administrated 0.25~1.2 mg/kg prednisolone daily to maintain parasite production in feces. Male Mongolian gerbils (TUM/MON strain) of 6–9 weeks were purchased from Charles Rivers (Wilmington, MA) and used as described below.

## Reagents

Dafachronic acids were synthesized as reported (*Basu et al., 2015*; *Mahanti et al., 2014*; *Sharma et al., 2009*). Methylprednisolone acetate (MPA) injection suspension (Zoetis, Parsippany, NJ), ketoconazole (Sigma, St. Louis, MO), 8-Br-cGMP (Tocris, Minneapolis, MN), and ivermectin (Merial Limited, Duluth, GA) were purchased as cited. Solvents for lipid extraction and liquid chromatography (toluene, hexanes, isopropanol, methanol, water, and acetonitrile) were purchased from Fisher Scientific (Waltham, MA). Eluent additives for liquid chromatography (LC)–mass spectrometry (MS) (formic acid or NH4Ac) were purchased from Sigma (St. Louis, MO). COS-7 and Sf9 cell lines were purchased from ATCC and are STR certified and mycoplasma-free.

## *S. stercoralis* culture

The wild-type (UPD strain) of *S. stercoralis* was maintained in purpose-bred dogs (Oak Hill Genetics, Ewing, IL) and FL stages of *S. stercoralis* were prepared from coroculture of the dog feces as described (*Lok, 2007*). Briefly, postparasitic worms at the noted stages were obtained using the Baermann technique (*Lok, 2007*) from daily collected dog feces (for PP-L1/L2); 24 hr coroculture at 18°C (for FL-L3); 72 hr coroculture at 21°C (for FL-adults and PFL-L1); and 7-day coroculture at 25°C (for L3i). The parasitic stages of *S. stercoralis* were acquired from Mongolian gerbils (Charles Rivers, Wilmington, MA) as described (*Lok, 2007*; *Stoltzfus et al., 2012*). To obtain L3+, gerbils infected with 10,000 L3i larvae were sacrificed 3-day postinfection. After removing the intestinal tract, parenteral tissues were minced and L3+ were collected by the Baermann technique at 37°C. To collect P-adults, intestinal L1–L3, and L3a, gerbils were hyperinfected by an initial inoculation of 4000 L3i and administration of 2 mg MPA, followed by weekly injections of 2 mg MPA for 4 weeks. After sacrificing the gerbils, P-females and intestinal larval stages were collected by hanging the small and large intestines in cylinders filled with saline water supplemented with 0.5 mg/ml gentamicin (Sigma, St. Louis, MO) at 37°C, which allows the parasites to migrate out of the tissues and settle down at bottom of the cylinders. The parasites were then collected and purified by manually removing each stage under a dissection scope. Collected parasites were then washed five times with M9 buffer containing an antibiotic cocktail containing 100 units penicillin, 100 µg/ml streptomycin (Thermo Fisher, Waltham, MA), and 50 µg/ml gentamicin (Sigma, St. Louis, MO) and stored in −80°C for lipid extraction.

## Lipid fractionation and ligand identification

Endogenous DAF-12 ligands were purified from the lipid fractions of PP-L3 as outlined (*Figure 2—figure supplement 1A*). Briefly, ~ 5 million PP-L3s were obtained from ~100 kg dog feces as described above. The worms were pooled and sonicated in 0.9% NaCl, from which crude lipids were extracted by the Folch method (*Folch et al., 1957*). The crude lipids were dissolved in toluene and 0.1% acetic acid and passed through a Sep-Pak Silica SPE columns (Waters, Milford, MA), followed by sequential elution with hexanes, 30% isopropanol in hexanes, and finally methanol, all of which contains 0.1% acetic acids. The lipids in the 30% isopropanol fraction were then fractionated with semipreparative high-performance liquid chromatography (HPLC) with a C18 column (Luna 5 µm C18 100 Å, 250 × 10 mm, Phenomenex, Torrance, CA) at a flow rate of 3.5 ml/min. The mobile phases were water (A) and acetonitrile (B), both containing 0.1% formic acid. The following gradient was run for a total of 70 min: 0–15 min with 50–100% (B); 15–70 min with 100% (B). Eluted fractions were collected every minute and dried under nitrogen gas and dissolved in ethanol.

HPLC fractions containing endogenous *Ss*-DAF-12 ligands were first identified using a cell-based reporter assay (*Wang et al., 2009*). Briefly, COS-7 cells (ATCC, Manassas, VA) cultured in DMEM (Thermo Fisher, Waltham, MA) supplemented with 10% fetal bovine Serum (FBS) (Gemini Bio, West Sacramento, CA) were cotransfected with plasmids expressing the *S. stercoralis* DAF-12 ligand-binding domain fused in frame with the DNA binding domain from *C. elegans* DAF-12 (pCMX-Ce*Ss*-DAF-12, to increase expression efficiency) (*Wang et al., 2009*), the *lit-1 kinase* DAF-12 reporter plasmid (pNL3.1-DAF-12RE-lit1, fused to Nanoluc luciferase) and the control reporter plasmid (pGL4.53, *pgk* promoter fused to firefly luciferase). The cells were split into 384-well plates and treated with 10 nM Δ7-DA or the HPLC fractions. After 24 hr, Nanoluc and firefly luciferase activities were measured by Nano-Glo Dual-luciferase kits (Promega, Madison, WI) on a Victor V plate reader (Perkin Elmer, Waltham, MA). Relative light units (RLUs) were then calculated by normalizing the Nanoluc luciferase activity to the firefly luciferase activity.

To characterize the chemical identity of the *Ss*-DAF-12 ligand, isolated HPLC fractions were subjected to ultra-performance liquid chromatography tandem with mass spectrometry (UPLC–MS) using a Shimadzu LC instrument (Shimadzu Scientific Instruments, Inc, Columbia, MD) in tandem with a Sciex 6500 Triple Quad mass spectrometer (AB Sciex LLC, Framingham, MA). The HPLC fractions were diluted in methanol containing 100 nM chenodeoxycholic acid-$^2$H$_4$ (d4-CDCA, Sigma, St. Louis, MO) as an internal control of retention times (RTs) between different runs. The lipids were then loaded onto a C18 column (Kinetex 1.3 μm C18 100 Å, LC Column 50 × 2.1 mm, Phenomenex, Torrance, CA) at a 0.4 ml/min flow rate. The mobile phase consisted of (A) water:acetonitrile (9:1) and (B) acetonitrile:water (99:1), both containing 2 mM NH4Ac. The following gradient was run for a total of 8 min: 60–80% of (B) over 0–2 min; 80% (B) over 2–4 min; 80–99% (B) over 4–6 min; and 99% (B) over 6–8 min. Using negative selective ion monitoring mode, Δ4-DA and Δ7-DA were analyzed at *m/z* 413 (RT = 1.89 min) and *m/z* 413 (RT = 2.08 min), respectively; Δ1,7-DA was analyzed at *m/z* 411 (RT = 1.95 min). The identity of the peaks was determined by comparing to standards that were run in parallel with the lipid fractions.

## Quantitative analysis of Δ7-DA levels in *S. stercoralis*

*S. stercoralis* at different developmental stages were prepared as described above and sample preparation is summarized in *Table 2*. For the environmental postparasitic stages (PP-L1, FL-L3, FL-adults, PFL-L1, and L3i), worms were lysed by sonication in 0.9% NaCl. For the host-derived parasitic stages (L3+, P-adult, and intestinal L1-L3a) and cGMP-treated L3i, worms were lysed in digestion buffer (2 mM Tris–HCl, pH 8.0, 1 mg/ml Proteinase K, and 0.25% SDS) at 55°C for 2 hr. SDS was then precipitated and removed by adding a saturating amount of KCl followed by centrifugation at 10,000 × *g* for 5 min. For quantification, 100 nM Δ7-DA was added as a standard to lipid fractions from stages that do not have endogenous Δ7-DA (listed in *Table 1*) and were processed in parallel with other samples. Lysates from parasite samples and standards were extracted by the Folch method and lipids were dried under nitrogen. Δ7-DA in the lipids was then derived to Δ7-DA-picolylamine (Δ7-DA-PA) for analysis by incubating the lipids with 3.3 mM of triphenylphosphine, 3.3 mM 2,2′-dipyridyl disulfide,

**Table 2.** Sample preparation for Δ7-DA quantification in *S. stercoralis*.

| Parasite sample | Amount | Lysis method | Standard preparation |
|---|---|---|---|
| FL-L1/L2 | 200,000 | Sonication | 100 nM Δ7-DA compound spiked in 200,000 FL-L1/L2 |
| FL-L3 | 50,000 | Sonication | 100 nM Δ7-DA compound spiked in 50,000 FL-L3 |
| FL-adult | 5000 | Sonication | 100 nM Δ7-DA compound spiked in 5000 FL-adults |
| PFL-L1 | 200,000 | Sonication | 100 nM Δ7-DA compound spiked in 200,000 PFL-L1 |
| L3i | 50,000 | Sonication | 100 nM Δ7-DA compound spiked in 50,000 L3i |
| L3+ | 1000 | Proteinase K | 100 nM Δ7-DA compound spiked in 1000 L3+ |
| P-adult | 500 | Proteinase K | 100 nM Δ7-DA compound spiked in 500 FL-adults |
| Intestinal L1-L3a | 5000 | Proteinase K | 100 nM Δ7-DA compound spiked in 5000 Int-larvae |
| cGMP-treated L3i | 1000 | Proteinase K | 100 nM Δ7-DA compound spiked in 1000 L3i |

and 333 ng/µl 2-picolylamine at 60°C for 20 min (*Li et al., 2013*). The Δ7-DA-PA was then analyzed by UPLC–MS. The UPLC method was same as described above and Δ7-DA-PA compounds were detected in positive multiple reaction monitoring (MRM) mode with $m/z$ transition 505 → 487. DA quantification was then performed with the software MultiQuant (AB Sciex LLC, Framingham, MA) by comparing Δ7-DA-PA peak areas in the parasite samples with those of 100 nM Δ7-DA standards.

## Biosynthesis of Δ7-DA from dietary cholesterol in *S. stercoralis*

PP-L1 stages of *S. stercoralis* were prepared from dog feces as described above and cultured on NGM plates modified for metabolic tracing of isotope-labeled cholesterol supplemented in the diet. Briefly, HB101 bacteria were cultured overnight in cholesterol-free medium (10 g/l ether-extracted peptone in DMEM) and prepared as 10× concentrates supplemented with 140 µM naturally labeled (i.e., $^{12}$C) or $^{13}$C$_3$ (Cambridge Isotope Laboratories, Tewksbury, MA) labeled cholesterol. The bacteria concentrates were then spotted on cholesterol-free NGM plates (3% agarose in S-basal) to form bacterial lawns. Approximately 10,000 PP-L1 larvae were cultured on the plates at 18°C for 24 hr and collected and analyzed as described above for Δ7-DA-PA-$^{13}$C$_3$ levels by UPLC–MS in the positive MRM mode with $m/z$ transition 508 → 490.

## In vitro enzyme activity assays

Candidate enzyme genes (*Stoltzfus et al., 2012*) were cloned and each gene expressed in Sf9 cells (ATCC, Manassas, VA), which are cultured with Sf 900 III medium (Thermo Fisher, Waltham, MA). A human CYP450 oxidoreductase (hOR) was coexpressed to facilitate electron transport required for the enzyme activities (*Motola et al., 2006*). Enzyme coding sequences were tethered with a C-terminal HA tag in pFastBac Dual baculoviral vectors (Invitrogen, Waltham, MA) using Ph or p10 expression cassettes for hOR or other enzymes, respectively. Baculovirus was prepared according to manufacturer's instructions followed by infection of Sf9 cells. For DAF-36 and DHS-16 homologs, the microsomes expressing the enzymes were purified from the infected Sf9 cells with a Microsome Isolation Kit (Abcam, Cambridge, MA). Microsomes were incubated with an NADPH regeneration system (Promega, Madison, WI) in presence or absence of 100 µM substrate compounds at 37°C for 16 hr. For CYP activities, infected Sf9 cells were incubated with 10 µM substrate compound at 28°C for 24 hr. The substrate compounds for DAF-36, DHS-16 and CYPs were cholesterol-$^2$H$_7$, lathosterol-$^2$H$_7$ (Avanti Polar Lipids, Inc, Alabaster, AL), and lathosterone (Steraloids, Inc, Newport, RI), respectively. Lipids were then extracted from microsomes (for DAF-36s and DHS-16s) or Sf9 cells (for CYPs) by the Folch method and loaded onto a C18 column (Kinetex 1.3 µm C18 100 Å, LC Column 50 × 2.1 mm, Phenomenex, Torrance, CA) at a 0.4 ml/min flow rate. The mobile phase consisted of (A) water:acetonitrile (9:1) and (B) acetonitrile:water (99:1), both containing 2 mM NH4Ac. For measuring the activities of the DAF-36 and DHS-16 homologs, the following gradient was run for a total of 8 min: 80–99% (B) for 0–2 min; 99% (B) for 2–8 min. Enzyme products were detected in positive MRM mode with $m/z$ transition 374 → 109 (7-dehydrocholesterol-$^2$H$_7$) and 392 → 109 (lathosterone-$^2$H$_7$). For CYP activities, Δ7-DA was analyzed by the UPLC–MS methods described above.

## L3i activation assay

Reactivation of L3i developmental arrest was assayed by monitoring feeding activity as described (*Albarqi et al., 2016*; *Cheong et al., 2021*). Briefly, L3i larvae prepared as described above were suspended in M9 buffer at 100 larvae/96 well. The larvae were treated with 0.5 mM 8-Br-cGMP (Torcis, Minneapolis, MN), 1 µM Δ7-DA and/or 25 µM of ketoconazole (Sigma, St. Louis, MO) at 37°C for 22 hr. To visualize the feeding behaviors, worms were incubated with 200 µg/ml of Alexa 594 fluorescent dye (Thermofisher, Waltham, MA) at 37°C for another 3 hr, washed and observed under fluorescent microscope. Worms with internal red fluorescence were scored as feeding larvae. For Δ7-DA levels, worms were suspended in M9 buffer at 4000 larvae/ml and treated for the indicated times. Δ7-DA levels were then analyzed by UPLC–MS as described above.

## QPCR

QPCR was performed by SYBR green method as reported (*Wang et al., 2015*). Briefly, 10,000–20,000 L3i larvae were pelleted and RNA extracted with RNA-STAT60 reagent (Amsbio, Cambridge, MA). Total RNA was digested with Turbo-DNAase (Ambion Inc, Austin, TX) and purified with RNeasy Mini

**Table 3.** Oligo sequences used in this study.

| | Sequence | Description |
|---|---|---|
| Forward | GGCATCACCATACAAAACAG | *Ss-cyp22a9* wild-type allele genotyping |
| Reverse | TTTGTATGAGGAGGGTTGTG | |
| Forward | GGCATCACCATACAAAACAG | *Ss-cyp22a9* KO allele genotyping |
| Reverse | CATCACATTCATCAAAAGTCCACT | |
| Forward | TCCTGGCCAGTGCTAATGTTATT | *Ss-cyp22a9* qPCR |
| Reverse | CTATTTGGACGGGATGAGAAGACT | |
| Forward | TGGTGCATGGCCGTTCTTA | *Ss-18SRNA* qPCR |
| Reverse | CTCGCTCGTTATCGGAATCAA | |
| Forward | GCTGGGGACTTATGGACAGG gttttagagctagaaatagcaag | sgRNA expression plasmid |
| Reverse | /5phos/CATTGTATTGGATGGCAATC | targeting *Ss-cyp22a9* |

Kits (Qiagen, Germantown, MD) to remove genomic DNA. Following reverse transcription, *Ss-cyp22a9* and *Ss-18S rRNA* levels were analyzed by QPCR (primer sequences in *Table 3*) and presented as relative mRNA levels (*Ss-cyp22a9/Ss-18S rRNA*).

## Immunoblotting

Western immunoblotting was performed following standard protocol. Sf9 cells expressing *Ss*-CYP450 enzymes were pelleted and lysed in Laemmli sample buffer by sonication followed by 5 min in a boiling water bath and 2 min on ice. The samples were then analyzed by 10% SDS–PAGE transferred to nitrocellulose membrane, and immunoblotted with the mouse anti-HA antibody [HA.C5] (Abcam, Cambridge, MA) followed by HRP-conjugated anti-mouse antibody (Abcam, Cambridge, MA). The membranes were then imaged by ImageQuant LAS4000 luminescent Image Analyzer (GE Healthcare, Chicago, IL) following an incubation with ECL chemiluminescent substrate reagent kit (Thermo Fisher, Waltham, MA).

## CRISPR gene disruption

To knock out expression of the gene encoding the DAF-9 ortholog, *Ss-cyp22a9*, CRISPR methods developed specifically for *S. stercoralis* were employed as outlined (*Figure 5—figure supplement 1*) and previously reported (*Cheong et al., 2021*; *Gang et al., 2017*; *Lok, 2019*). Briefly, an sgRNA targeting *Ss-cyp22a9* was designed by GPP sgRNA Designer (Broad Institute) and CHOPCHOP (http://chopchop.cbu.uib.no/). Top candidates with 5'-(N)₁₈GG-3' formats from both designs were compared to the *S. stercoralis* genome (https://parasite.wormbase.org/Strongyloides_stercoralis_prjeb528/Info/Index/) for off-targeting analysis. The sequence 5'-GCTGGGGACTTATGGACAGG-3' was selected as only targeting the *Ss-cyp22a9* locus within the entire *S. stercoralis* genome and cloned into the gRNA vector pML60 (*Gang et al., 2017*). *Ss-unc-22* sgRNA (gift from Dr. Ellisa A. Hallem, UCLA) was used as a control. CRISPR gene editing was accomplished using both homologous directed repair (HDR) and nonhomologous end joining (NHEJ) and vectors as previously published (*Gang et al., 2017*; *Lok, 2019*). For HDR, a homology repair template was made by sewing ~500 bp homology arms flanking the Cas9 cutting site at the *Ss-cyp22a9* locus with a GFP expressing cassette (*Sr-eef-1p::GFP::era*, from pPV529) (*Figure 5—figure supplement 1C*) by fusion PCR. At least 50 FL females of *S. stercoralis* were microinjected with plasmids expressing sgRNA (60 ng/µl), Cas9 (pPV540, 20 ng/µl), and the homology repair template (10 ng/µl). Injected females were then placed with adult males in the fecal culture as described (*Cheong et al., 2021*; *Gang et al., 2017*; *Lok, 2019*). The F1 progenies from these matings were collected after 7-day fecal culture at 25°C with the Baermann technique. The GFP-positive L3i larvae from the F1 progenies were manually picked and treated with 0.5 mM 8-Br-cGMP or 1 µM Δ7-DA in M9 buffer at 37°C for 22 hr. The feeding behaviors were then visualized as described above and genotyped individually by single worm PCR with the primers listed in *Table 3*.

In a second set of experiments, NHEJ CRISPR was used to produce enough *Ss-cyp22a9* knockout worms for detection of endogenous Δ7-DA. As described above, 50 FL females of *S. stercoralis* were microinjected with plasmids expressing Cas9 (pPV540, 20 ng/µl) and sgRNAs against *Ss-cyp22a9* or *Ss-unc-22* (60 ng/µl). Following a fecal culture with FL males at 25°C for 7 days, the resulting L3i larvae were collected by the Baermann technique and treated by 0.5 mM 8-Br-cGMP in M9 buffer for 3 days. Levels of Δ7-DA were analyzed by UPLC–MS as described above.

### Gerbil models of *S. stercoralis* infection

Uncomplicated and hyperinfection cases of strongyloidiasis were established in Mongolian gerbils (Charles Rivers, Wilmington, MA) as described (*Nolan et al., 1993*). For uncomplicated strongyloidiasis, gerbils were subcutaneously injected with 1000 L3i *S. stercoralis* larvae. On day 21 post the infection, the gerbils were switched to drinking water containing 5% sucrose (vehicle) or 50 µM Δ7-DA in 5% sucrose. We have found that adding a small amount of sucrose does not affect infection, but ensures animals drink appropriate amounts of drug. Fecal larval numbers were monitored biweekly for the next 14 days. To establish hyperinfection, gerbils were infected similarly but MPA (2 mg/gerbil) was injected subcutaneously weekly from the time of infection. After 10-day postinfection to allow time for the hyperinfection to occur, Δ7-DA treatment was begun as noted above. Ivermectin (Merial Limited, Duluth, GA) was intraperitoneally injected at 300 µg/kg on day 21 postinfection. Since Δ7-DA is rapidly turned over in vivo (*Patton et al., 2018*), administering it at an earlier time point ensured an adequate amount was delivered. Animals were observed daily for irreversible hyperinfection symptoms that indicate imminent lethality, which include lethargy, hunched back, scruffy hair, lack of activity, and reduced drinking. Animals with such symptoms were sacrificed and autopsy performed to determine end-point parasite numbers as described below.

### Parasite burden analysis

Parasite numbers in gerbils were determined as described with modifications (*Lok, 2007*; *Nolan et al., 1993*; *Stoltzfus et al., 2012*). For fecal larvae outputs, infected gerbils were housed in wire bottom cages with a layer of dampened paper towel for 8 hr. Feces (15–20 droppings) were collected and macerated in 0.9% NaCl. The fecal suspension was then filtered through a layer of cheese cloth and an aliquot of the suspension was spotted on NGM plates. After drying, plates were incubated at 37°C for 10 min and living larvae, which leave traces on the plates, were counted and normalized to the feces weight as larvae per gram feces. To determine end-point parasite burden, intestines were cut longitudinally and incubated in 0.9% NaCl supplemented with an antibiotic cocktail (100 units/ml penicillin, 100 µg/ml streptomycin and 50 µg/ml gentamicin) at 37°C for 2–3 hr. The suspension was then filtered through a layer of cheese cloth and an aliquot of the suspension was spotted on NGM plates. Following drying and 10 min incubation at 37°C, parasitic adults and intestinal L1–L3a larvae were counted. To determine parasite burden of hyperinfected L3+, parenteral tissues were dissected, minced and larvae collected by the Baermann method were counted on watch glasses.

### Data analysis

Data were plotted and analyzed by the indicated statistical tests using GraphPad Prism 8 software to obtain p values. When multiple comparisons were involved, pairwise statistical analyses were first performed to obtain p values and then adjusted by the false discovery rate method using the p.adjust function in R to obtain *q* values (https://www.R-project.org/).

## Acknowledgements

We thank members of the Mango/Kliewer lab and Dr. Jeffrey McDonald (UT Southwestern) for fruitful discussions; Dr. Elissa Hallem (UCLA) for providing CRISPR plasmids; Adeiye Pilgrim (Emory) for assistance with cloning. This work was supported by the National Institutes of Health (grant AI105856 to JBL, DJM, and SAK, GM141088 to TQ, and AI050886 to JBL), the Robert A Welch Foundation (grants I-1275 to DJM, I-1558 to SAK, and I-2010-20190330 to TQ), UT Southwestern Eugene McDermott Scholarship (TQ), and the Howard Hughes Medical Institute (DJM).

# Additional information

## Funding

| Funder | Grant reference number | Author |
|---|---|---|
| National Institutes of Health | AI105856 | James B Lok<br>Steven A Kliewer<br>David J Mangelsdorf |
| National Institutes of Health | GM141088 | Tian Qin |
| National Institutes of Health | AI050886 | James B Lok |
| Welch Foundation | I-1275 | David J Mangelsdorf |
| Welch Foundation | I-1558 | Steven A Kliewer |
| Welch Foundation | I-2010-20190330 | Tian Qin |
| Howard Hughes Medical Institute | | David J Mangelsdorf |

The funders had no role in study design, data collection, and interpretation, or the decision to submit the work for publication.

## Author contributions

Zhu Wang, Conceptualization, Data curation, Formal analysis, Investigation, Methodology, Validation, Writing – original draft, Writing – review and editing; Mi Cheong Cheong, Investigation, Methodology; Jet Tsien, Investigation, Resources, Optimized the synthesis and provided dafachronic acids; Heping Deng, Investigation, Optimized the synthesis and provided dafachronic acids, Resources; Tian Qin, Funding acquisition, Investigation, Methodology, Resources, Supervision, Validation, Writing – review and editing, Optimized the synthesis and provided dafachronic acids; Jonathan DC Stoltzfus, Investigation, Methodology, Writing – review and editing; Tegegn G Jaleta, Resources, Generated parasite material; Xinshe Li, Resources, Generated parasite material; James B Lok, Conceptualization, Formal analysis, Funding acquisition, Methodology, Project administration, Resources, Supervision, Writing – review and editing; Steven A Kliewer, Conceptualization, Formal analysis, Investigation, Project administration, Supervision, Writing – original draft, Writing – review and editing; David J Mangelsdorf, Conceptualization, Funding acquisition, Methodology, Project administration, Supervision, Validation, Writing – original draft, Writing – review and editing

## Author ORCIDs

Zhu Wang http://orcid.org/0000-0003-0768-0988
Jet Tsien http://orcid.org/0000-0002-2052-8051
Jonathan DC Stoltzfus http://orcid.org/0000-0002-4006-5306
Steven A Kliewer http://orcid.org/0000-0001-5161-641X
David J Mangelsdorf http://orcid.org/0000-0002-4355-0796

## Ethics

All animal experiments were approved by the University of Texas Southwestern Medical Center Institutional Animal Care and Use Committee (IACUC) and are listed under animal protocol numbers 2016-101500 and 2018-102369. All protocols, as well as routine husbandry care of the animals, were conducted in strict accordance with the Guide for the Care and Use of Laboratory Animals of the National Institutes of Health (National Research Council, 2011).

## Decision letter and Author response

Decision letter https://doi.org/10.7554/eLife.73535.sa1
Author response https://doi.org/10.7554/eLife.73535.sa2

## Additional files

### Supplementary files
• Transparent reporting form

### Data availability
All data are presented in the manuscript; source data files have been provided for Figure 4—figure supplement 1 and Figure 5B.

The following previously published datasets were used:

| Author(s) | Year | Dataset title | Dataset URL | Database and Identifier |
|---|---|---|---|---|
| Hunt VL, Tsai IJ, Coghlan A, Reid AJ, Holroyd N, Foth BJ, Tracey A, Cotton JA, Stanley EJ, Beasley H, Bennett HM, Brooks K, Harsha B, Kajitani R, Kulkarni A, Harbecke D, Nagayasu E, Nichol S, Ogura Y, Quail MA, Randle N, Xia D, Brattig NW, Soblik H, Ribeiro DM, Sanchez-Flores A, Hayashi T, Itoh T, Denver DR, Grant W, Stoltzfus JD, Lok JB, Murayama H, Wastling J, Streit A, Kikuchi T, Viney M, Berriman M | 2016 | *Strongyloides stercoralis* | https://parasite.wormbase.org/Strongyloides_stercoralis_prjeb528/Info/Index/ | WormBase ParaSite, GCA_000947215.1 |

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
