## [Editor Report]

This work reveals the pathway by which an important human parasite synthesizes a nuclear hormone receptor ligand critical for progression through its life cycle and demonstrates the potential therapeutic implications of perturbing this pathway. The experiments are insightfully and expertly conceived, designed and executed, and the data support the conclusions. This manuscript will be of general interest to parasitologists, nematode biologists, and those studying transcriptional regulatory networks governed by ligand-gated nuclear receptors.

---

## [Decision Letter]

**Decision letter after peer review:**

Thank you for submitting your article "Characterization of the Endogenous DAF-12 Ligand and Its Use as an Anthelmintic Agent in *Strongyloides stercoralis*" for consideration by *eLife*. Your article has been reviewed by 2 peer reviewers, and the evaluation has been overseen by Phillip Newmark as Reviewing Editor and Dominique Soldati-Favre as the Senior Editor. The following individuals involved in review of your submission have agreed to reveal their identity: Mostafa Zamanian (Reviewer #1); Keith Yamamoto (Reviewer #2).

Essential revisions:

1) As noted by reviewer #2 (see their public review), the Results section claims that unliganded Ss-DAF-12 is required for production of infectious L3i larvae, but this is not demonstrated here. The authors could either provide evidence demonstrating the requirement of unliganded Ss-DAF-12 for L3i production, or they could summarize briefly in the Results section how they established this requirement, citing their PNAS 2021 publication at that point (as well as in the Discussion, where it is currently first cited in this context).

---

## [Author Response]

Essential revisions:1) As noted by reviewer #2 (see their public review), the Results section claims that unliganded Ss-DAF-12 is required for production of infectious L3i larvae, but this is not demonstrated here. The authors could either provide evidence demonstrating the requirement of unliganded Ss-DAF-12 for L3i production, or they could summarize briefly in the Results section how they established this requirement, citing their PNAS 2021 publication at that point (as well as in the Discussion, where it is currently first cited in this context).

We appreciate the reviewer’s comment and to address this we have updated the text to state in the Introduction (lines 98-100) and Results (lines 166-167) that the findings highlighted in our previous PNAS paper did demonstrate that *Ss*-DAF-12 is required for L3i production. This was also already stated in the first paragraph of the Discussion (lines 342-343). In addition, we note that the present study also demonstrates this requirement by showing that the absence of the DAF-12 ligand (and thus unliganded *Ss*-DAF-12) is necessary for L3i formation and the presence of the ligand is necessary and sufficient for reactivation of L3i arrest and reproductive development.